# Optimizing vaccine allocation for COVID-19 vaccines shows the potential role of single-dose vaccination

Laura Matrajt [1✉], Julia Eaton [2], Tiffany Leung [1], Dobromir Dimitrov[1,3], Joshua T. Schiffer[1,4,5], David A. Swan[1] & Holly Janes[1]

Most COVID-19 vaccines require two doses, however with limited vaccine supply, policymakers are considering single-dose vaccination as an alternative strategy. Using a mathematical model combined with optimization algorithms, we determined optimal allocation strategies with one and two doses of vaccine under various degrees of viral transmission. Under low transmission, we show that the optimal allocation of vaccine vitally depends on the single-dose efficacy. With high single-dose efficacy, single-dose vaccination is optimal, preventing up to 22% more deaths than a strategy prioritizing two-dose vaccination for older adults. With low or moderate single-dose efficacy, mixed vaccination campaigns with complete coverage of older adults are optimal. However, with modest or high transmission, vaccinating older adults first with two doses is best, preventing up to 41% more deaths than a single-dose vaccination given across all adult populations. Our work suggests that it is imperative to determine the efficacy and durability of single-dose vaccines, as mixed or single-dose vaccination campaigns may have the potential to contain the pandemic much more quickly.

[1] Vaccine and Infectious Disease Division, Fred Hutchinson Cancer Research Center, Seattle, WA, USA. [2] School of Interdisciplinary Arts and Sciences, University of Washington, Tacoma, WA, USA. [3] Department of Applied Mathematics, University of Washington, Seattle, WA, USA. [4] Department of Medicine, University of Washington, Seattle, WA, USA. [5] Clinical Research Division, Fred Hutchinson Cancer Research Center, Seattle, WA, USA. ✉email: laurama@fredhutch.org

COVID-19 has killed over 2,400,000 people worldwide as of February 21, 2021[1]. With several vaccines proven highly efficacious (estimated at 94.1%, 95%, 82%, and 91.6% for Moderna, Pfizer, AstraZeneca, and Sputnik V, respectively) against COVID-19[2–5], hopes are high that a return to normal life can soon be possible. Twenty other vaccines are currently in phase 3 clinical trials[6]. Most of these vaccines require two doses given at least 3 weeks apart[7]. Since a large proportion of the global population needs to be vaccinated to reduce transmission and mortality, vaccine supply shortage will be inevitable in the first few months of vaccine availability. Even in high-income countries, which have secured the largest quantities of vaccine, supply will be highly insufficient initially[8]. This situation could be far worse in low- and middle-income countries (LMIC), where vaccine supplies might arrive at later times and in smaller quantities, with limited vaccine supply for LMIC risking the public and economic health of those populations, as well as that of the global population[9].

Most of the current vaccine prioritization schedules use two-dose deployments[10], but the logistics of a two-dose vaccination campaign, which ensures a second dose for those who have already received one dose, are challenging especially in the context of limited vaccine supply and shelf-life[11]. In previous outbreaks of other infectious diseases, fractional dosing, where people receive less than the recommended dosage of vaccine, has been successfully utilized as a way to stretch vaccine supply. A single-dose campaign of the killed oral cholera vaccine (which also requires two doses) was deployed in a recent cholera outbreak in Zambia, where the population was vaccinated with one dose and, months later, high-risk individuals were offered a second dose[12]. In 2016, in response to a yellow fever outbreak in Angola, Uganda and the Democratic Republic of Congo, a vaccination campaign with one fifth of the recommended dosage of the yellow fever vaccine[13] was successfully carried out[13,14]. If sufficiently effective, single-dose COVID-19 vaccination is attractive for several reasons: it is easier to implement logistically, potentially less costly, and a larger proportion of the population could be vaccinated in a fixed amount of time, thereby potentially reaching herd immunity levels and allowing resumption of key community activities (e.g., reopening schools, restaurants, gyms, etc.) more rapidly[15–17]. This may be especially true if COVID-19 vaccines not only reduce disease but also prevent infection and hence onward transmission; these are open questions and data are still emerging on the full spectrum of vaccine effects[3,18,19]. However, the success of a COVID-19 single-dose vaccination campaign depends on the protection acquired after one dose of vaccine. There is an intrinsic trade-off with using single-dose vaccination campaigns achieving greater vaccine coverage, in exchange for a potentially lower level of protection and/or less durable protection.

In this work, we addressed two questions of public-health importance: (1) Who should be vaccinated first? and (2) How many doses should individuals receive? Utilizing mathematical models combined with optimization algorithms, we determined the optimal allocation of available vaccine doses under a variety of assumptions, and at levels of vaccine efficacy consistent with estimates from phase 3 efficacy trials. We independently minimized five metrics of infection and disease burden: cumulative infections, cumulative symptomatic infections, cumulative deaths, and peak non-ICU and ICU hospitalizations. The last two metrics were chosen as a way to evaluate healthcare system burden. We showed that mixed vaccination strategies in which some age groups receive one dose while others receive two doses can achieve the greatest reductions in these metrics under fixed vaccine quantities. Further, our results suggest that the optimal vaccination strategy depends on the relative efficacy of single-versus full-dose vaccination; on the full spectrum of vaccine effects; on the number of vaccine doses available; and on the speed of vaccine rollout and the intensity of background transmission. This highlights the critical importance of continued research to define the level of efficacy conferred by a single vaccine dose and to evaluate efficacy against not only COVID-19 disease, but SARS-CoV-2 infection and secondary transmission.

## Results

As it is expected that vaccine supplies will ramp up considerably over the second half of 2021 and into 2022, and because there is considerable uncertainty around durability of vaccine protection, we focused on the first few months of vaccine availability and set 6 months for the duration of our simulations. We built upon our previous model of SARS-CoV-2 transmission and vaccination[20]. Briefly, we developed a deterministic age-structured mathematical model with the population of Washington state (7.6 million people) and US demographics divided into 16 age groups with contact rates given in ref. [21], adjusted for reciprocity (Supplementary Fig. 1). To perform the vaccine optimization, we collapsed the 16 age-groups into 5 vaccination age-groups: 0–19, 20–49, 50–64, 65–74, and those 75 and older, aligned with vaccination groups currently considered by the Centers for Disease Control and Prevention (CDC)[22]. We assumed that at the beginning of our simulations, 20% of the population has pre-existing immunity (through infection)[23] distributed proportionally to the population, and the SARS-CoV-2 prevalence (number of current SARS-CoV-2 active infections) was 0.1% of the population[24] (alternative scenarios: 20% pre-existing immunity with different distributions across the population, 10% pre-existing immunity and 0.05% or 0.3% prevalence, see Sensitivity Analysis and Supplementary Information (SI)). We assumed that asymptomatic infections are 75% as infectious as symptomatic infections (alternative scenario: 30% as infectious, see Sensitivity Analysis and SI) and confer complete immunity upon recovery, over our time horizon of 6 months. Further, we assumed that both naturally acquired immunity and vaccine-induced immunity (one- and two-dose) do not wane during our 6 month time horizon.

We considered a baseline basic reproductive number (denoted $R_0$) $R_0 = 3$ (alternative scenario $R_0 = 4$, Sensitivity Analysis) with four levels of social distancing interventions that would affect the contact rates and therefore SARS-CoV-2 transmission, such that, in combination with the assumed level of pre-existing immunity, resulted in effective reproductive numbers (defined as the average number of secondary cases per infectious case in a population made up of both susceptible and non-susceptible hosts, denoted $R_{eff}$) of $R_{eff} = 1.1$ (observed in WA state in January 2021[25]), 1.3, 1.5, and 2.4, respectively, at the beginning of our simulations (Supplementary Table 3 and Supplementary Fig. 2 and SI). Of course, as vaccination and the epidemic process progress, the $R_{eff}$ will change. We evaluated five metrics of disease and healthcare burden: cumulative infections, cumulative symptomatic infections, cumulative deaths, maximum number of non-ICU hospitalizations and maximum number of ICU-hospitalizations. State goals for limiting hospital and ICU beds occupied by COVID-19 patients[26,27] were used for result interpretation.

**Modeling the vaccine effects.** Ongoing phase 3 COVID-19 vaccine trials evaluate vaccine efficacy against laboratory-confirmed COVID-19, or the multiplicative reduction in per-exposure risk of disease, which we denote by $VE_{DIS}$. We considered a leaky vaccine (that is, a vaccine that confers partial protection to all vaccinated individuals) that can have three effects on vaccinated individuals[28]: to reduce the probability of acquiring a SARS-CoV-

2 infection (measured by $VE_{SUS}$), reduce the probability of developing COVID-19 symptoms after infection (measured by $VE_{SYMP}$), or reduce the infectiousness of vaccinated individuals upon infection (measured by $VE_I$, Supplementary Fig. 3A).

Given the efficacy data on two-dose COVID-19 vaccines to date[2,3,29,30], we considered a main scenario with $VE_{DIS}$ = 90%. As many combinations of $VE_{SUS}$ and $VE_{SYMP}$ can result in the same $VE_{DIS}$ (Supplementary Fig. S3B), and in advance of definitive data on $VE_{SUS}$ or $VE_{SYMP}$ for COVID-19 vaccines, we considered three different vaccine profiles that yield $VE_{DIS}$ = 90%: a vaccine effect mediated by $VE_{SUS}$ only, a vaccine effect mediated by $VE_{SYMP}$ only, and a vaccine effect that is a combination of $VE_{SUS}$ and $VE_{SYMP}$ (Supplementary Fig. 3B and Supplementary Table 1). In the absence of data on the vaccine effect on infectiousness, we took a conservative approach and assumed $VE_I$ = 0 (alternative scenario, $VE_I$ = 70%, Sensitivity Analysis). Given the limited data to-date regarding the efficacy of single-dose vaccination of vaccines intended to be given in a two-dose schedule[4,31], we considered three "single-dose efficacy" (SDE) scenarios, under our main scenario where two-dose $VE_{DIS}$ = 90%: low SDE, whereby the single-dose vaccine confers low efficacy against COVID-19 disease ($VE_{DIS_1}$ =18%); moderate SDE with $VE_{DIS_1}$ = 45%; and high SDE with $VE_{DIS_1}$ = 72%; corresponding to 20%, 50%, and 80% of the 90% efficacy of the full two-dose regimen, respectively. The efficacy of single-dose vaccination against infection and symptoms were assumed to be reduced proportionally. All vaccine effects were assumed to take effect immediately following each vaccine dose and to remain constant after the last vaccine dose over the time horizon of 6 months. For two-dose vaccination, we explicitly modeled vaccination campaigns with the first dose, followed by vaccination campaigns with the second dose, so that individuals receiving two doses had the protection conferred by single-dose vaccination in the inter-vaccination period.

**Modeling vaccination campaigns.** We considered the distribution of enough vaccine doses to cover from 10% to 50% of the population with a single dose. We simulated vaccination campaigns delivering 150,000 (150K) vaccine doses per week, with a maximum of 50% of the population vaccinated with a single dose of vaccine over $a \sim$ 6-month period (our time horizon). This matches current vaccination plans in the US[32]. An alternative scenario with 300K vaccine doses per week was also explored (Sensitivity Analysis). These are roughly twice and four times, respectively, the vaccination rate experienced in the US during the 2009 H1N1 influenza pandemic[33].

Throughout the text, we refer to vaccine coverage as the amount of vaccine available to cover a percentage of the population with one dose of vaccine. For each vaccination scenario and disease metric, we denote the optimal strategy as the allocation that was found to minimize a given disease metric, as determined by our optimization routine. We compared the optimal strategy to two other strategies: a pro-rata strategy in which one-dose vaccination is rolled out across all adult age groups proportional to their population size (this strategy models an allocation in which all adults are eligible to be vaccinated and we assumed that all age groups are equally likely to be vaccinated); and a high-risk strategy in which two-dose vaccination is allocated to the oldest age groups first and then to younger age groups in decreasing order as vaccine availability permits (similar to the current prioritization strategy in the US[34]). For example, with 50% vaccine coverage, under the pro-rata strategy 66.5% of each age group (excluding children) would receive a single dose of vaccine, and under the high-risk strategy all of those aged 65 and older and 44% of those aged 50 to 64 years

would receive two doses of vaccine (Supplementary Fig. 4). We also compared the optimal strategy to a pragmatic strategy, where all adults aged 65 and older receive two doses of vaccine and all other adults receive a single dose as coverage permits. The results for this strategy were nearly identical to those from the high-risk strategy, so we present only the latter set of results. In order to perform the optimization, we implemented the vaccination campaigns in an identical way for all the allocation strategies considered: the older age group in a particular strategy is vaccinated first and we moved sequentially in decreasing order across the vaccine groups (full details of the implementation are given in the SI).

To assess parameter uncertainty, for each of the strategies compared and for each outcome, we ran the model with 1,000 different parameter sets representing the uncertainty surrounding those parameters we believe would be more likely to affect our results, and removed top and bottom 2.5% of the simulations to calculate uncertainty intervals (denoted below as 95% UI) reflecting the uncertainty in the outcomes arising from uncertainty in the parameter estimates (Uncertainty analysis, SI). A full description of the model and the methods can be found in the SI.

In the main scenario, we considered $R_{eff}$ = 1.1, a vaccine mediated both by $VE_{SUS}$ and $VE_{SYMP}$, so that $VE_{DIS}$ = 90% after two doses with $VE_{SUS}$ = 70% and $VE_{SYMP}$ = 66%, a vaccination campaign with 150K doses per week, focusing on minimizing COVID-19 deaths.

**Results assuming low background viral transmission.** In this section, we assumed that vaccination started under strong social distancing interventions, resulting in a low background transmission and an $R_{eff}$ = 1.1 or $R_{eff}$ = 1.3, as experienced in WA state in January 2021[25]. For low SDE combined with low vaccination coverage (enough vaccine to cover up to 20% of the population with a single dose), the optimal strategy allocated two doses of vaccine to the high-transmission group (adults aged 20–49 in our model) and the high-risk groups (adults aged 65 and older, Figs. 1A, D and 2A). The optimal strategy averted up to 37% more deaths compared to the pro-rata strategy ($R_{eff}$ =1.3 and 20% coverage with a single dose, optimal: 54% (95% UI: 51–55) vs. pro-rata: 16% (95% UI: 12–19) deaths averted compared with no vaccination) and 12% more deaths compared to the high-risk strategy ($R_{eff}$ =1.1 and 10% coverage, optimal: 42% (95% UI: 35–45) vs. high-risk: 30% (95% UI: 27–31)), Fig. 3A, D. Even at low coverage, the optimal strategy to minimize deaths also resulted in a significant reduction in overall transmission and hospitalizations (Fig. 2 and Supplementary Fig. 5). As coverage increased, the optimal strategy prioritized coverage in older adults with two doses of vaccine (Fig. 1G, J, and M), averting over 34% more deaths than the pro-rata strategy, with a maximum of 45% more deaths averted ($R_{eff}$ =1.3, 40 and 50% coverage, optimal: 68% (95% UI: 60–70) vs. pro-rata: 22% (95% UI: 17–25)), Fig. 3D.

For moderate SDE, the optimal vaccine allocation prioritized the high-risk groups (65 and older) with mixed vaccination strategies (one and two doses of vaccine) to increase coverage of these groups when few doses were available, and with two doses of vaccine as coverage increased (Fig. 1B, E, H, K, and N), averting up to 18% more deaths than the pro-rata strategy ($R_{eff}$ = 1.3, 50% coverage, optimal: 71% (95% UI: 63–72) vs. pro-rata: 53% (95% UI: 45–55)). However, the optimal strategy provided only a modest gain when compared to the high-risk strategy, averting an additional 13% of deaths at low coverage ($R_{eff}$ = 1.1, 10% coverage, optimal: 43% (95% UI: 36–47) vs. high-risk: 30% (95% UI: 28–32)), and averted a similar number of deaths for higher levels of coverage (30% or higher), Fig. 3B, E.

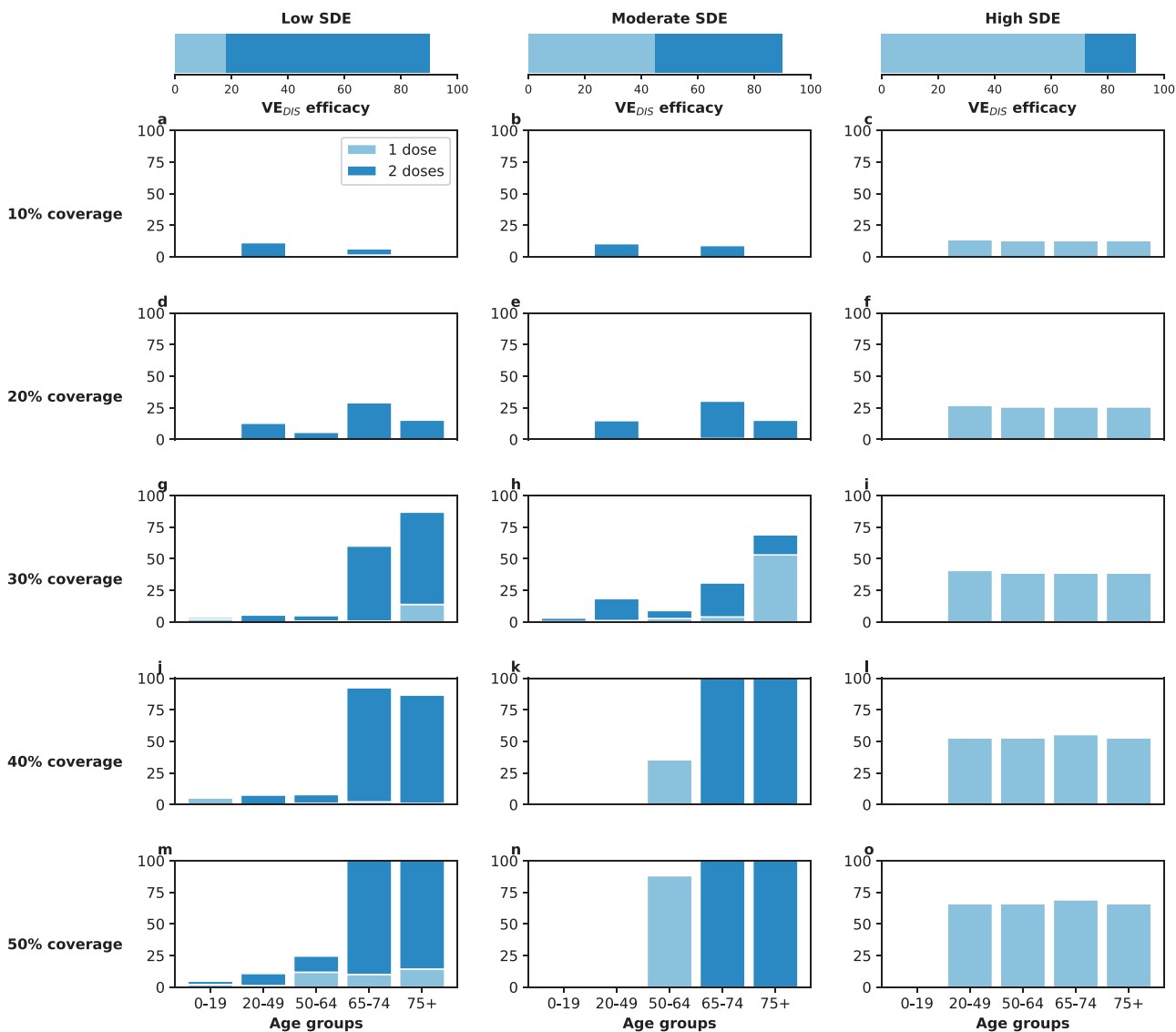

**Fig. 1 Optimal vaccine allocation strategies for minimizing deaths for different vaccination coverages.** For each plot, the bars represent the percentage of each age group vaccinated with a single dose (light blue) and two doses (dark blue) when there is enough vaccine to cover 10% (**a–c**), 20% (**d–f**), 30% (**g–i**), 40% (**j–l**) or 50% (**m–o**) of the population with a single dose. The columns correspond to assumptions that the single-dose efficacy (SDE) is low (left column, $VE_{DIS_1} = 18\%$), moderate (center column, $VE_{DIS_1} = 45\%$) or high (right column, $VE_{DIS_1} = 72\%$), corresponding 20%, 50%, or 80% of the full vaccine efficacy, $VE_{DIS} = 90\%$ assumed following two doses of vaccine, respectively. Here, we assumed an effective reproductive number $R_{eff} = 1.1$.

In stark contrast, if the vaccine was highly efficacious after one dose, then, for all levels of coverage, the optimal strategy was almost identical to the the pro-rata strategy (Fig. 1C, F, I, L, and O), and it averted up to 22% more deaths than the high-risk strategy ($R_{eff} = 1.1$, 10% coverage, optimal: 53% (95% UI: 45–57) vs. high-risk: 31% (95% UI: 28–32)), Fig. 3C, F. While all strategies averted similar number of deaths for high coverage (40 and 50% coverage), the optimal strategy and the pro-rata strategy had the advantage that they significantly slowed down viral transmission and as a result, reduced the overall prevalence of infections (Fig. 4C, F).

**Results assuming moderate or high background viral transmission**. We next considered that vaccination started under less strict social distancing interventions, resulting in moderate or high background transmission with an $R_{eff} = 1.5$ or $R_{eff} = 2.4$, as experienced in WA state early in the pandemic and in November 2020, respectively[25]. Here, the optimal vaccine allocation was

always to directly protect those at highest risk (older adults): for low and moderate SDE or for a high SDE combined with high coverage, it was optimal to vaccinate these groups with two doses of vaccine (Fig. 5A, B, D, E, G–O), and if the SDE was high and few doses were available (≤20%) then it was optimal to vaccinate them with a single dose of vaccine (≤20%, Fig. 5C, F) to boost the coverage of that group. If the SDE was low or moderate, the optimal strategy was almost identical to the high-risk strategy, averting up to 47% more deaths than the pro-rata strategy (low SDE, $R_{eff} = 1.5$, 30% coverage, optimal: 59% (95% UI: 48–63) vs. pro-rata:12% (95% UI: 11–15)), Fig. 3G, H, J, K. For high SDE, the gain from optimizing vaccine allocation to improve high-risk strategy was relatively small, averting up to 12% more deaths ($R_{eff} = 1.5$, 10% coverage optimal: 41% (95% UI: 40–42) vs. high-risk: 29% (95% UI: 27–29)) but averted up to 23% more deaths than the pro-rata strategy ($R_{eff} = 2.4$, 10% coverage, optimal: 36% (95% UI: 35–37) vs. pro-rata: 13% (95% UI: 12–13) deaths averted). With moderate viral transmission, the optimal strategy to minimize deaths slightly reduced the overall prevalence

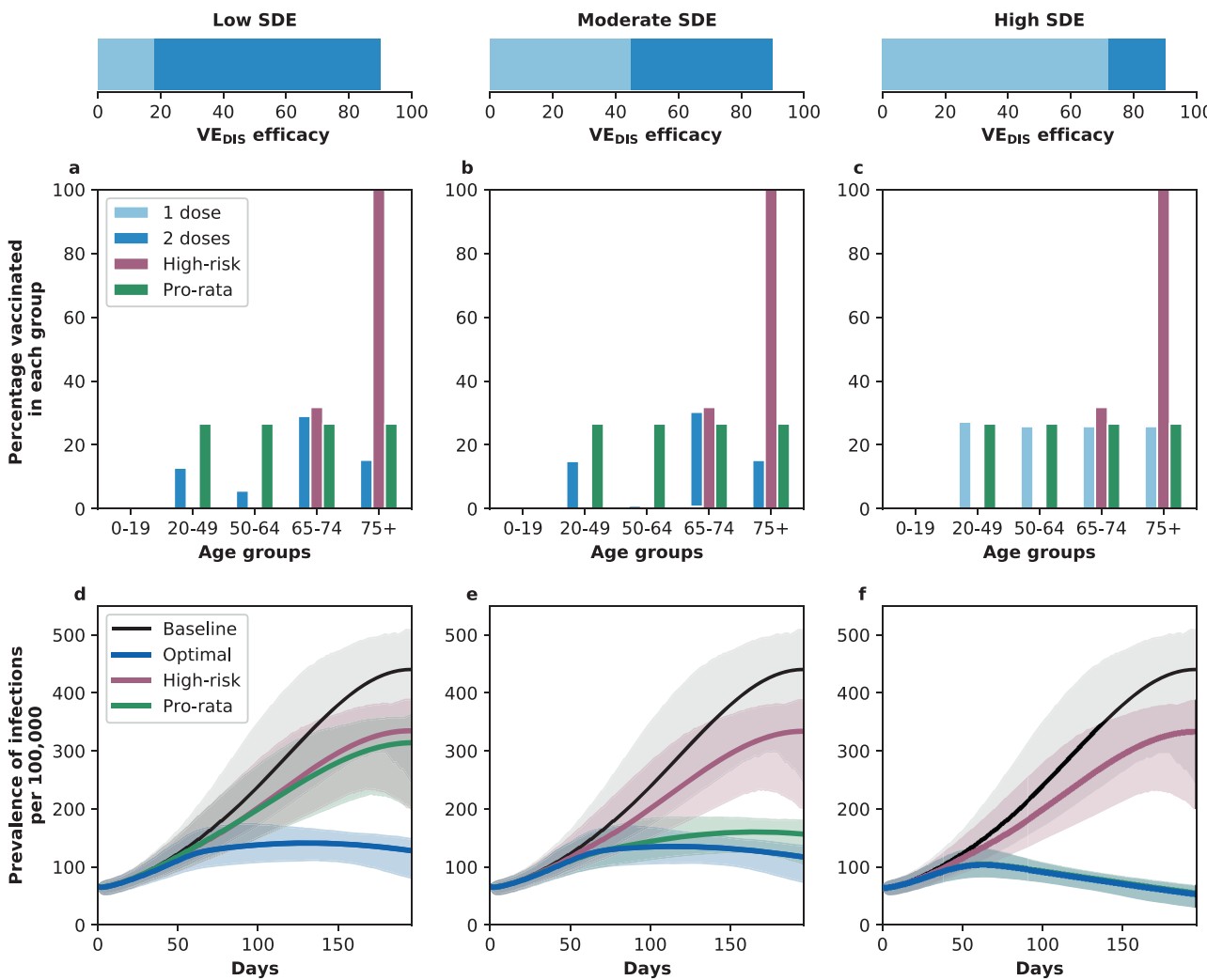

**Fig. 2 Allocation strategies and prevalence of active infections with enough vaccine for 20% coverage with a single dose. a–c** Optimal, pro-rata, and high-risk allocation strategies with 20% coverage. Optimal allocation strategy to minimize deaths (light and dark blue), high-risk (pink) and pro-rata (green) allocation strategies assuming enough vaccine to cover 20% of the population with a single dose (10% with two doses). Within each panel, the bars represent the percentage vaccinated in each vaccination group. **d–f** Prevalence of active infections. Prevalence of active infections (per 100,000) in the absence of vaccine (black), with the optimal allocation strategy to minimize deaths (blue), the high-risk strategy (pink) or the pro-rata strategy (green) with enough vaccine to cover 20% of the population with a single dose (10% with two doses). Shaded areas represent central 95% of 1000 simulations (uncertainty intervals, see SI for full details). Additional plots for the prevalence of non-ICU and ICU hospitalizations are shown in Supplementary Fig. 5. The columns correspond to assumptions that the single-dose efficacy (SDE) is low (left column, $VE_{DIS_1} = 18\%$), moderate (center column, $VE_{DIS_1} = 45\%$) or high (right column, $VE_{DIS_1} = 72\%$), corresponding 20%, 50%, or 80% of the full vaccine efficacy, $VE_{DIS} = 90\%$ assumed following two doses of vaccine, respectively. Here, we assumed an effective reproductive number $R_{eff} = 1.1$.

(regardless of the SDE). However, for high viral transmission, even with enough vaccine to cover 50% of the population, none of the strategies optimized to minimize deaths reduced transmission (Fig. 4G–L).

Overall, the optimal strategy outperformed the high-risk and the pro-rata strategies the most under low background transmission and low coverage. For all levels of viral transmission considered, regardless of the SDE, the epidemic advanced at a faster pace than the vaccination campaign, evidenced by the fact that the percentage of deaths averted plateaued at 40% ($R_{eff} = 1.1$), 30% coverage ($R_{eff} = 1.3, 1.5$) or even 20% coverage ($R_{eff} = 2.4$) Fig. 3).

**Different metrics have different optimal allocation strategies.** The other metrics we considered capture different disease and healthcare burden impacts of the pandemic, and minimizing each produced a different optimal allocation strategy.

When minimizing outcomes affecting transmission (total infections or total symptomatic infections) and low SDE, the optimal strategy allocated a mix of one and two doses across all age groups, Fig. 6 and S6A and D. In contrast, with moderate or high SDE, the optimal strategy allocated most of the available vaccine to the most active group (adults aged 20–49) with a single dose, Fig. 6 and Supplementary Fig. 6B, C, E, and F.

When minimizing peak non-ICU hospitalizations and for low or moderate SDE, the optimal strategy prioritized adults aged 65–74 with mixed vaccination strategies to boost coverage of this age group (Figs. 6 and Supplementary Fig. 6G, H). When minimizing ICU hospitalizations with a vaccine with low or moderate SDE, it was optimal to vaccinate the high-risk group (75 and older) at high coverage with two doses of vaccine, and all the other groups with mixed allocations of one and two doses, Fig. 6 and Supplementary Fig. 6J and K. If a single-dose vaccine was highly efficacious, then for all metrics minimizing severe

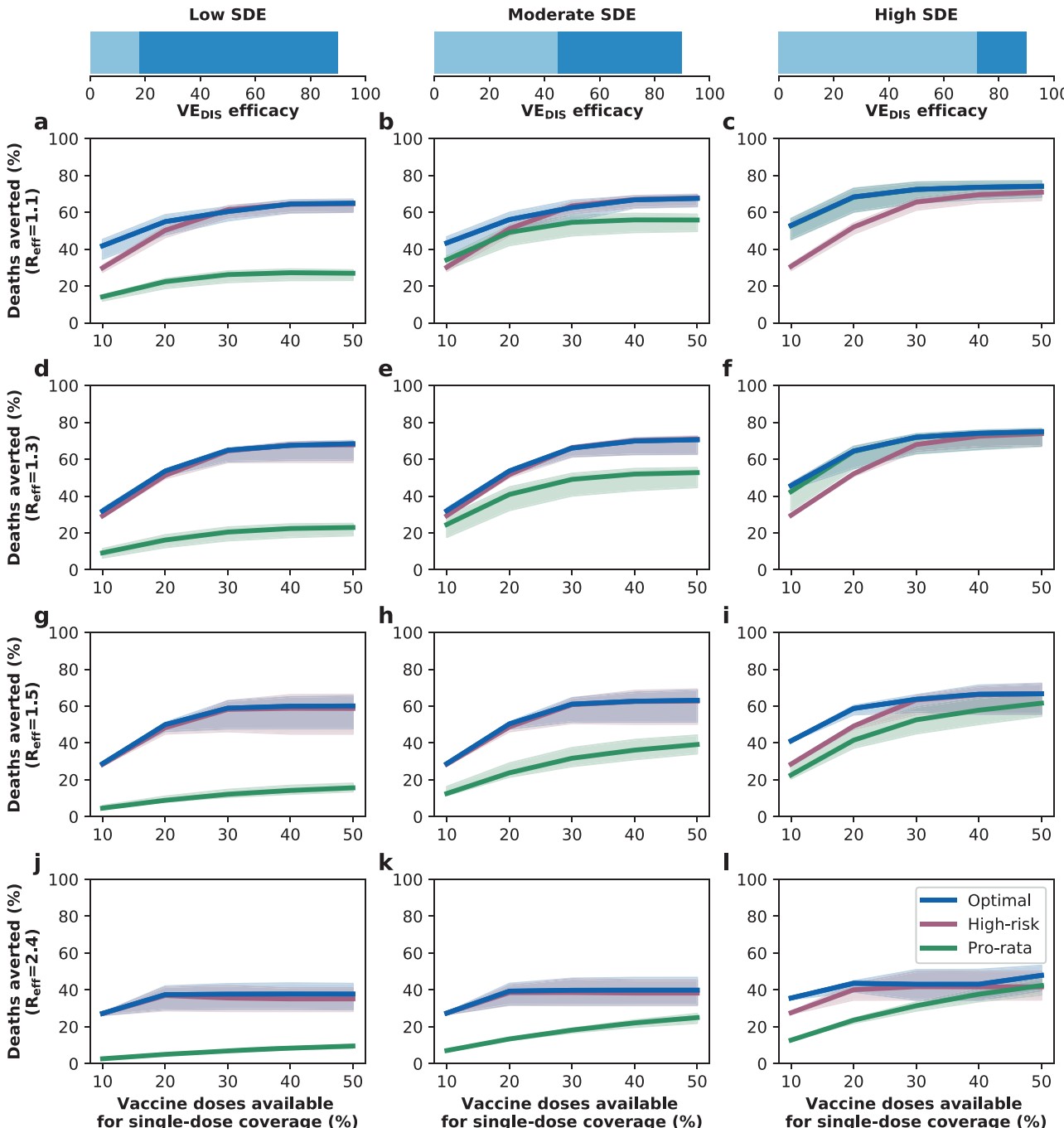

**Fig. 3 Percentage of deaths averted for different levels of SARS-CoV-2 transmission.** Percentage of deaths averted for the optimal allocation strategy to minimize deaths (blue), the high-risk strategy (pink) and the pro-rata strategy (green) with enough vaccine to cover 10–50% of the population with one dose. Each row represents a different level of SARS-CoV-2 transmission resulting in an effective reproductive number $R_{eff} = 1.1$ (**a–c**), 1.3 (**d–f**), 1.5 (**g–i**) or 2.4 (**j–l**). Shaded areas represent central 95% of 1000 simulations (uncertainty intervals, see SI for full details). The columns correspond to assumptions that the single-dose efficacy (SDE) is low (left column, $VE_{DIS_1} = 18\%$), moderate (center column, $VE_{DIS_1} = 45\%$) or high (right column, $VE_{DIS_1} = 72\%$), corresponding 20%, 50%, or 80% of the full vaccine efficacy, $VE_{DIS} = 90\%$ assumed following two doses of vaccine, respectively. Shaded areas represent central 95% of 1000 simulations (uncertainty intervals, see SI for full details).

disease (hospitalizations) or deaths and for nearly all levels of coverage, the optimal strategy was in fact the pro-rata strategy, (Fig. 6 and Supplementary Fig. 6I, L, and O). However, we noted that even in those scenarios where the optimal and the pro-rata strategies did not coincide, they averted the same number of deaths.

Interestingly, at this level of viral transmission ($R_{eff} = 1.1$), the peak hospitalizations (even in absence of vaccine) stayed below WA state desired thresholds (a maximum of 10% of general hospital beds occupied by COVID-19 patients and no overflow of the ICU beds, Fig. S5D–I). If viral transmission increased so that $R_{eff} = 1.3$, baseline hospitalizations were above the desired

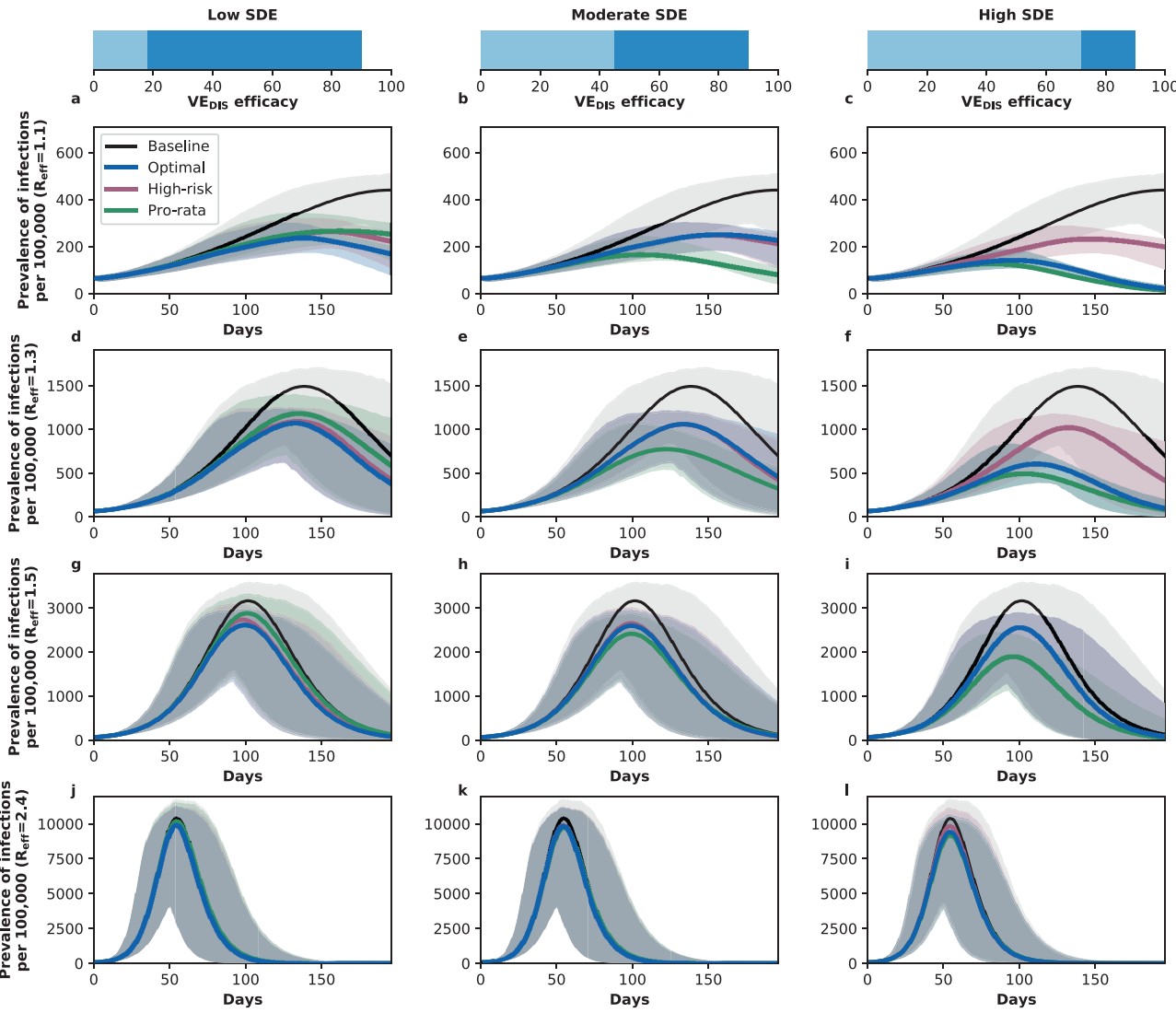

**Fig. 4 Prevalence of active infections (per 100,000) for different levels of background transmission.** Prevalence of infections (per 100,000) in the absence of vaccine (black), with the optimal allocation strategy to minimize deaths (blue), the high-risk strategy (pink) or the pro-rata strategy (green) with enough vaccine to cover 50% of the population with a single dose (25% with two doses). Each row represents a different level of SARS-CoV-2 transmission resulting in an effective reproductive number $R_{eff} = 1.1$ (**a–c**), 1.3 (**d–f**), 1.5 (**g–i**), or 2.4 (**j–l**). Shaded areas represent central 95% of 1000 simulations (uncertainty intervals, see SI for full details). The columns correspond to assumptions that the single-dose efficacy (SDE) is low (left column, $VE_{DIS_1} = 18\%$), moderate (center column, $VE_{DIS_1} = 45\%$) or high (right column, $VE_{DIS_1} = 72\%$), corresponding 20%, 50%, or 80% of the full vaccine efficacy, $VE_{DIS} = 90\%$ assumed following two doses of vaccine, respectively.

thresholds, and only the optimal strategy achieved Washington state goals of staying below these thresholds with as little as 20% vaccination coverage, irrespective of the SDE. For higher coverage, all strategies considered achieved this goal provided that the SDE was moderate or high (Fig. 7 and Supplementary Fig. 7). However, once $R_{eff} \geq 1.5$, the peak non-ICU and ICU hospitalizations for all strategies were higher than desired, even at 50% coverage (Supplementary Fig. 8).

A summary of the optimal allocation strategies for different combinations of SDE and background transmission for minimizing total infections, hospitalizations and deaths is given in Fig. 8.

**The vaccine profile shapes the optimal allocation strategy.** In this section we analyzed how different vaccine profiles affected the optimal allocation strategies. With a vaccine effect on COVID-19 disease mediated exclusively by a reduction in symptoms (high $VE_{SYMP}$), the optimal strategies for minimizing

deaths allocated two doses of vaccine to older adults (aged 65 and older) for direct protection (Fig. 9a–c), irrespective of the SDE. If the reduction in disease was mediated by both a reduction in symptoms and preventing infection, or predominantly by preventing infection ($VE_{SUS}$), then for low SDE it was still optimal to protect higher-risk groups with two doses (Fig. 9D, G). For moderate SDE, directly protecting the higher-risk groups was still optimal, with two doses if the vaccine was mediated by both effects (moderate $VE_{SYMP}$ and moderate $VE_{SUS}$) and with a single dose if it was exclusively mediated by preventing infection (Fig. 9E, H) However, if the SDE was high, the pro-rata strategy performed best (Fig. 9F, I).

A vaccine preventing only symptomatic disease has the potential to prevent only up to 67% (95% UI: 63–69) of deaths over 6 months compared to 78% (95% UI: 72–80) reduction in mortality if exclusively mediated by preventing infection (Fig. 10C, I).

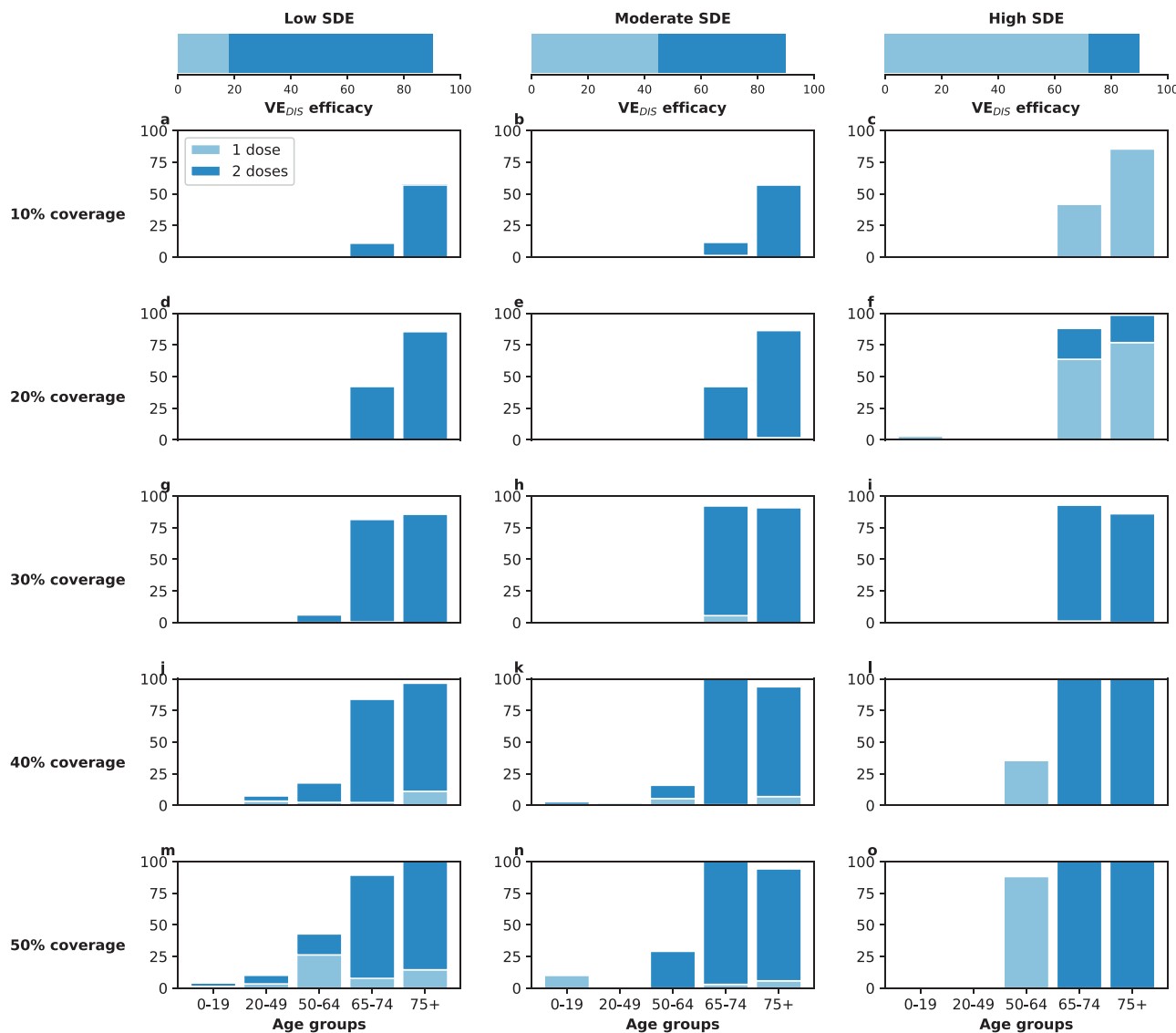

**Fig. 5 Optimal vaccine allocation strategies to minimize deaths with different levels of coverage, assuming an effective reproductive number $R_{eff}$ = 1.5.** For each plot, the bars represent the percentage of each age group vaccinated with a single dose (light blue) and two doses (dark blue) when there is enough vaccine to cover 10% (**a-c**), 20% (**d-f**), 30% (**g-i**), 40% (**j-l**) or 50% (**m-o**) of the population with a single dose. The columns correspond to assumptions that the single-dose efficacy (SDE) is low (left column, $VE_{DIS_1}$ = 18%), moderate (center column, $VE_{DIS_1}$ = 45%) or high (right column, $VE_{DIS_1}$ = 72%), corresponding 20%, 50%, or 80% of the full vaccine efficacy, $VE_{DIS}$ = 90% assumed following two doses of vaccine, respectively.

The vaccine efficacy profile had little effect on the cumulative deaths averted by the high-risk strategy (regardless of the SDE) but had a major effect on the pro-rata strategy: while this allocation performed very poorly if the vaccine was exclusively mediated by $VE_{SYMP}$ and low SDE, with 17% (95% UI: 15–19) deaths averted (compared with no vaccination) at 50% coverage, it was much more effective if the vaccine was mediated exclusively by $VE_{SUS}$ and had a high SDE, averting 78% (95% UI: 72–80) of the deaths for the same coverage (Fig. 10A, I).

Moderate protection against infection ($VE_{SUS}$) was important to all vaccination strategies to ensure reduction in transmission but especially for the optimal strategy (Supplementary Fig. 9). A vaccine acting exclusively by reducing symptoms had a smaller impact on the overall transmission with a maximum of 41% (95% UI: 31–48) cumulative infections averted (high SDE, 50% coverage), Fig. S9C) while a vaccine that reduced SARS-CoV-2 acquisition could, if

optimally allocated, averted a maximum of 88% (95% UI: 83–90) of cumulative infections (high SDE, 50% coverage), Supplementary Fig. 9I.

**Sensitivity analysis for asymptomatic infectiousness.** Owing to the wide range of estimates of the relative infectiousness of asymptomatic infections to symptomatic ones (ranging from 0.2 to 1[35]), we repeated our analysis assuming asymptomatic infections were 30% as infectious as symptomatic ones. The most noticeable differences were found when we analyzed the effect of different vaccine profiles. For low SDE, it was still optimal to directly protect high-risk groups, with two doses if the vaccine was mediated by reducing symptoms or if it was mediated by a combination of reducing symptoms and preventing infection, and with a single dose if it was mediated exclusively by preventing infection (Supplementary Fig. 10A, D, G). For moderate SDE,

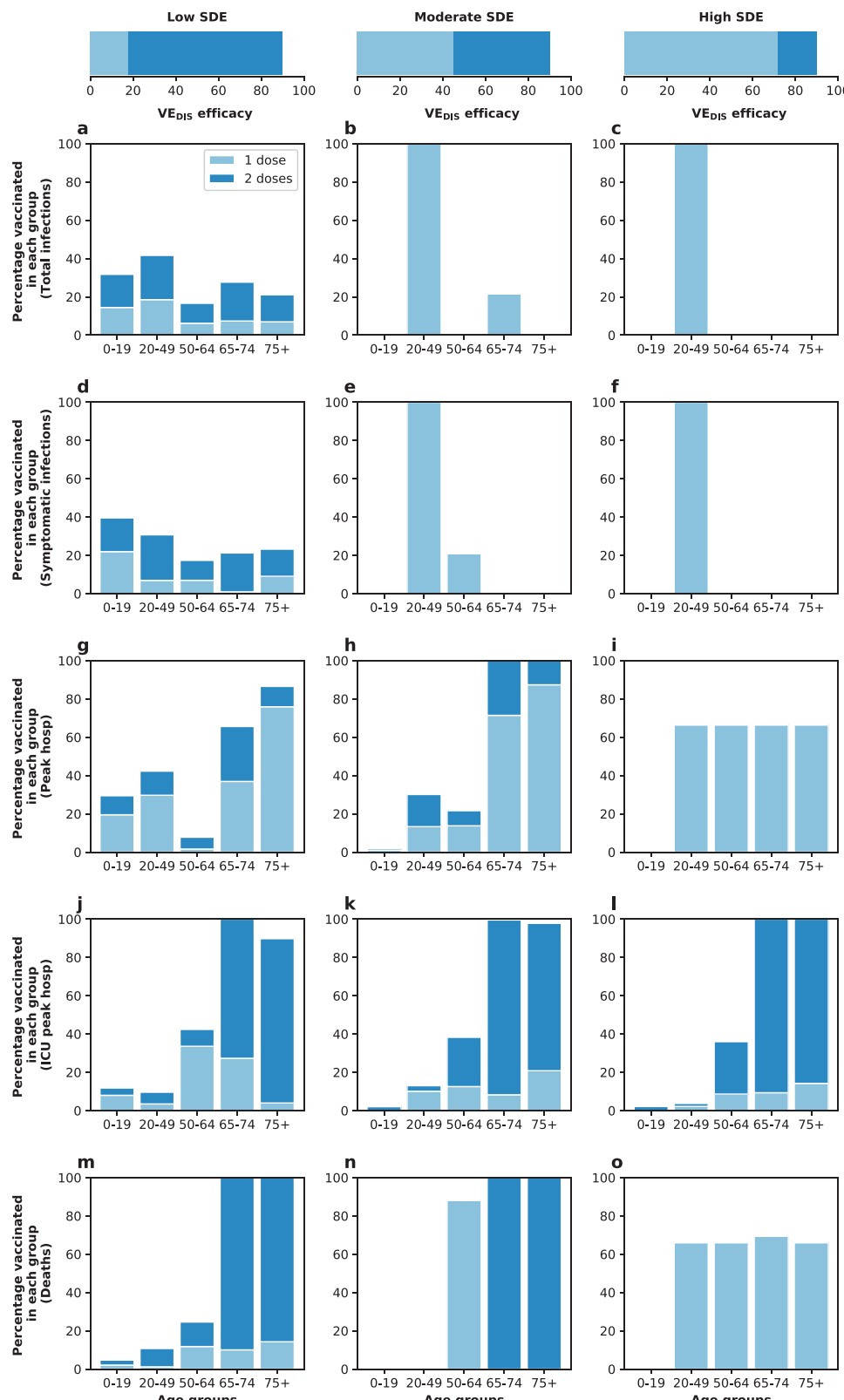

**Fig. 6 Optimal vaccine allocation strategies for different disease metrics with 50% coverage.** Optimal vaccine allocation assuming enough vaccine to cover 50% of the population with a single dose (25% with two doses). Each row represents a different disease metric minimized: cumulative infections (**a**–**c**), cumulative symptomatic infections (**d**–**f**), non-ICU peak hospitalizations (**g**–**i**), ICU hospitalizations (**j**–**l**) and total deaths (**m**–**o**). The columns correspond to assumptions that the single-dose efficacy (SDE) is low (left column, $VE_{DIS_1} = 18\%$), moderate (center column, $VE_{DIS_1} = 45\%$) or high (right column, $VE_{DIS_1} = 72\%$), corresponding 20%, 50%, or 80% of the full vaccine efficacy, $VE_{DIS} = 90\%$ assumed following two doses of vaccine, respectively. Here, we assumed an effective reproductive number $R_{eff} = 1.1$.

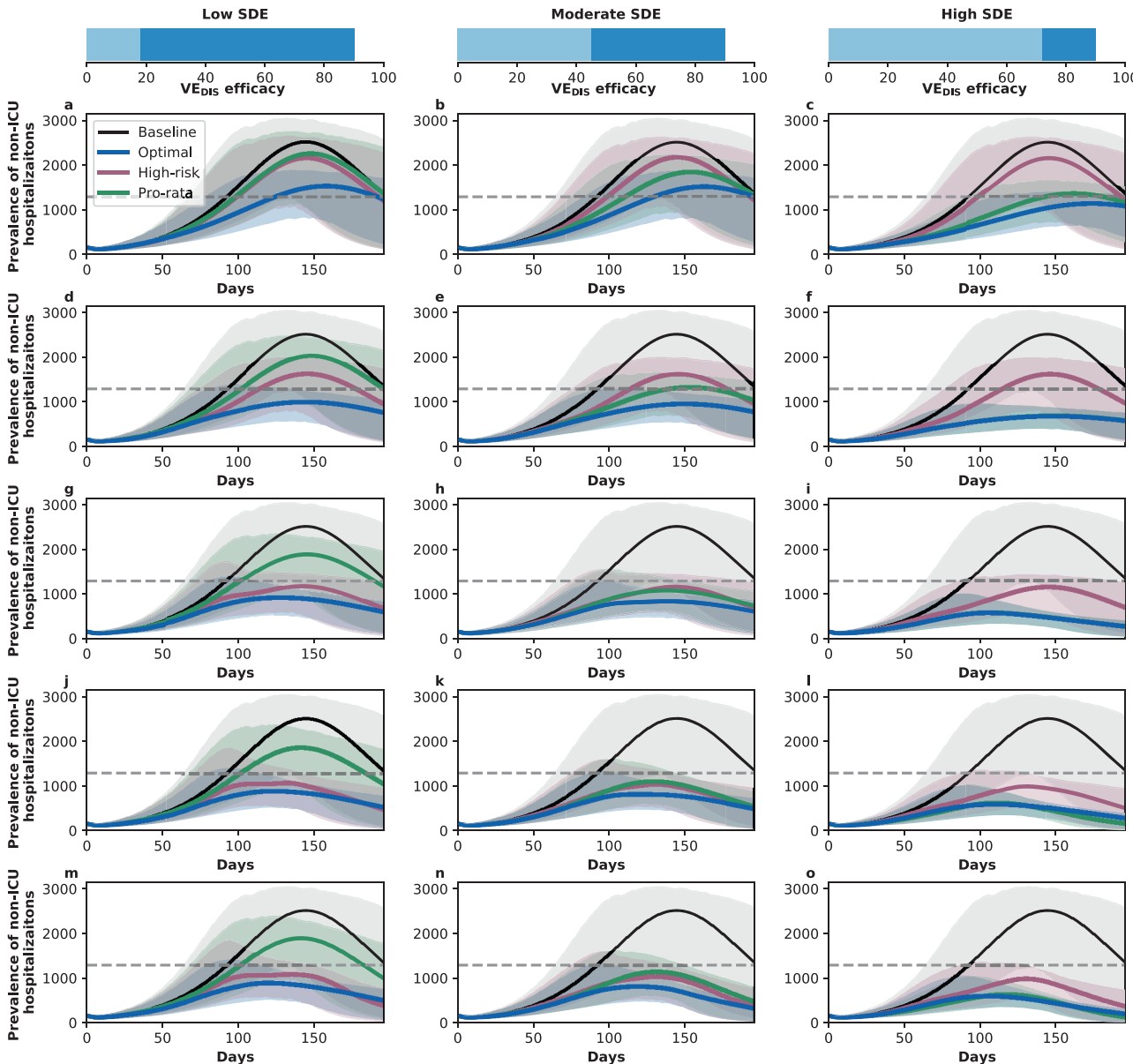

**Fig. 7 Prevalence of non-ICU hospitalizations with an effective reproductive number $R_{eff} = 1.3$.** Prevalence of non-ICU hospitalizations in absence of vaccine (black), with the optimal allocation strategy to minimize non-ICU hospitalizations (blue), the high-risk strategy (pink) or the pro-rata strategy (green). The gray dashed line indicates 10% occupancy of non-ICU beds in WA state. Each row corresponds to a different vaccination coverage, ranging from 10% (**a**–**c**) to 50% (**m**–**o**) coverage with a single dose. Shaded areas represent central 95% of 1000 simulations (uncertainty intervals, see SI for full details). The columns correspond to assumptions that the single-dose efficacy (SDE) is low (left column, $VE_{DIS_1} = 18\%$), moderate (center column, $VE_{DIS_1} = 45\%$) or high (right column, $VE_{DIS_1} = 72\%$), corresponding 20%, 50%, or 80% of the full vaccine efficacy, $VE_{DIS} = 90\%$ assumed following two doses of vaccine, respectively.

mixed allocations were optimal, with more use of single-dose vaccination with a vaccine exclusively mediated by preventing infection (Supplementary Fig. 10B, E, H). For high SDE, regardless of the vaccine profile, the pro-rata strategy was in fact, the optimal strategy (Supplementary Fig. 10C, F, I). As expected, if asymptomatic infections are considerably less infectious then a vaccine mediated exclusively by reducing symptomatic disease has a higher impact on the overall transmission, preventing as much as 81% (95% UI: 76–83) of total infections (compared to a maximum of 41% (95% UI: 31–48) total infections averted in the main scenario, Supplementary Figs. 9C and 11C.

**Sensitivity analysis for pre-existing immunity.** We repeated the main analysis assuming 10% of the population has pre-existing

immunity at beginning of vaccination. The results were very similar to those presented in the main scenario, with optimal vaccination strategies favoring single-dose campaigns if the SDE is high (Supplementary Fig. 12). In this scenario, the epidemic grows faster than in the main scenario and the assumed reduction in contacts presented in Supplementary Table 3 resulted in $R_{eff} = 1.3$. As a result, the projected impacts of different strategies are very similar to the ones presented above with $R_{eff} = 1.3$ with the high-risk strategy being optimal for low and moderate SDE when combined with high coverage and pro-rata strategy being optimal for high SDE irrespective of coverage (Fig. 3d–f).

**Sensitivity analysis for infection prevalence at the beginning of vaccination.** We investigated the effect of infection prevalence

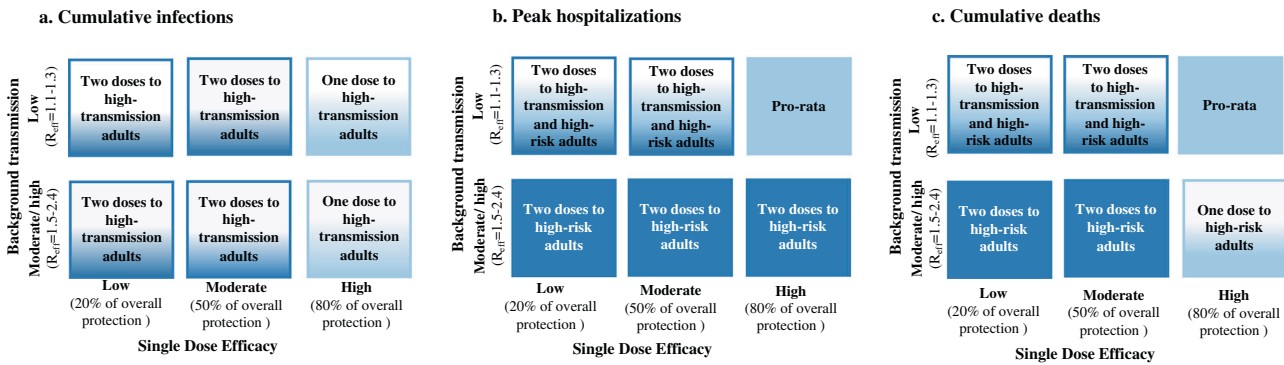

**Fig. 8 Lay summary of vaccine optimization assuming low vaccine coverage.** Each box represents a given single-dose vaccine efficacy and background transmission combination. Box color is light blue if the optimal strategy approximates pro-rata vaccine prioritization and dark blue if the optimal strategy approximates high-risk vaccine prioritization. A color blend is used when a hybrid approach is optimal. Text in the boxes describes the optimal approach. All results imply low vaccine coverage and 20% pre-existing immunity. Panels indicate optimal allocations for cumulative infections (**a**), peak hospitalizations (**b**) and cumulative deaths (**c**).

**Fig. 9 Optimal vaccine allocation to minimize deaths for different vaccine profiles with 50% coverage.** Optimal vaccine allocation for minimizing deaths assuming enough vaccine to cover 50% of the population with a single dose (25% with two doses). For each panel (**a**–**i**), the bars represent the total percentage of the population in each vaccination group to be vaccinated, split in those receiving a single dose (light blue) and those receiving two doses (dark blue). Each row represents a different breakdown of vaccine efficacy against disease after two doses $VE_{DIS} = 90\%$ as a function of the vaccine efficacy reducing susceptibility to infection, $VE_{SUS}$, and the vaccine efficacy reducing the probability of developing COVID-19 symptoms upon infection, $VE_{SYMP}$. Top row (**a**–**c**): $VE_{DIS}$ is exclusively mediated by a reduction in symptoms upon infection. Middle row (**d**–**f**): $VE_{DIS}$ is mediated by a combination of reduction in susceptibility to infection and reduction of symptoms upon infection. Bottom row (**g**–**i**): $VE_{DIS}$ is exclusively mediated by a reduction in susceptibility to infection. The columns correspond to assumptions that the single-dose efficacy (SDE) is low (left column, $VE_{DIS_1} = 18\%$), moderate (center column, $VE_{DIS_1} = 45\%$) or high (right column, $VE_{DIS_1} = 72\%$), corresponding 20%, 50%, or 80% of the full vaccine efficacy, $VE_{DIS} = 90\%$ assumed following two doses of vaccine, respectively. Here, we assumed an effective reproductive number $R_{eff} = 1.1$.

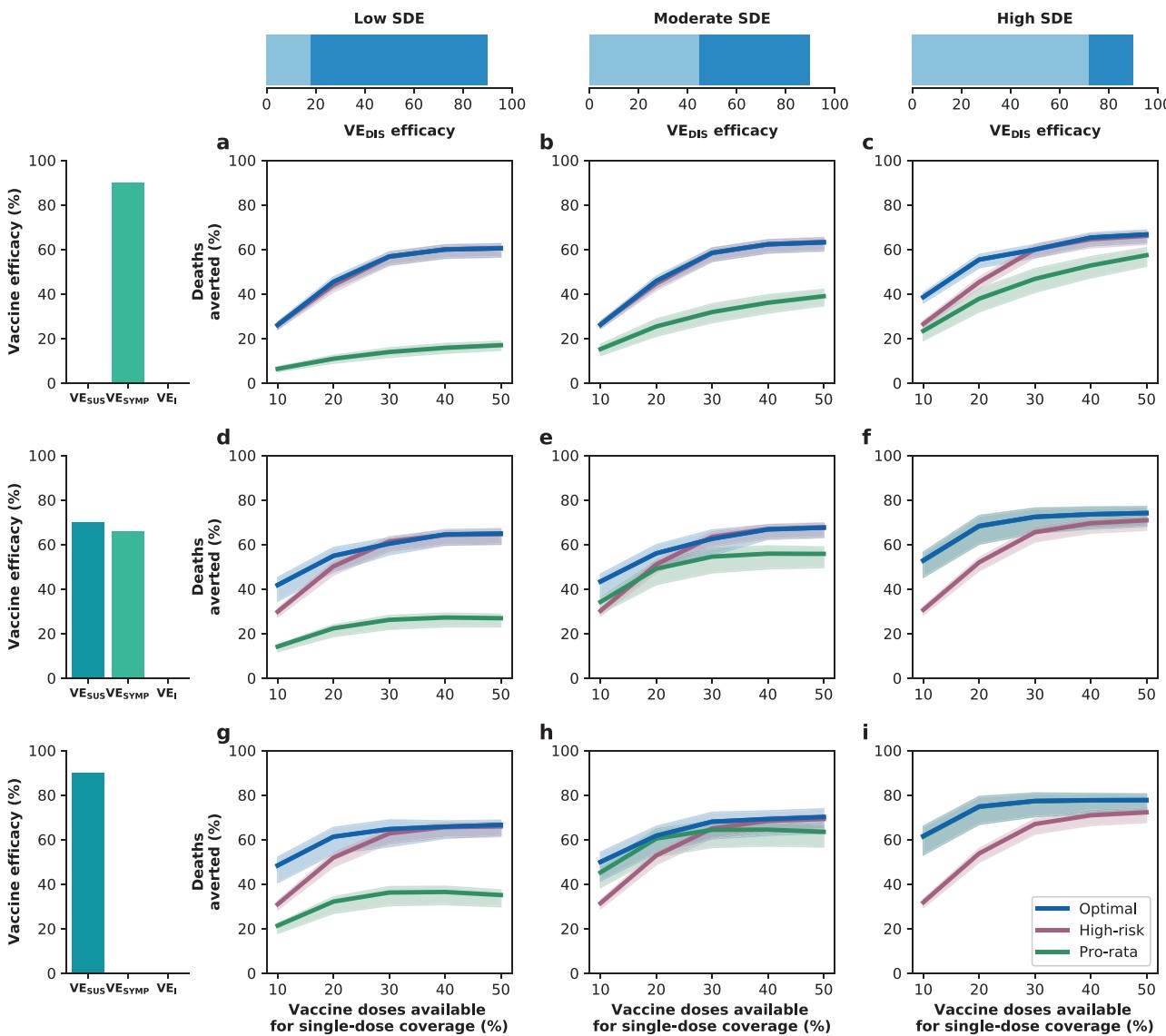

**Fig. 10 Percentage of deaths averted for different vaccine profiles.** Percentage of deaths averted for the optimal allocation strategy to minimize deaths (blue), the high-risk strategy (pink) and the pro-rata strategy (green) with enough vaccine to cover 10–50% of the population with one dose. Each row represents a different breakdown of vaccine efficacy against disease after two doses $VE_{DIS} = 90\%$ as a function of the vaccine efficacy reducing susceptibility to infection, $VE_{SUS}$, and the vaccine efficacy reducing the probability of developing COVID-19 symptoms upon infection, $VE_{SYMP}$. Top row (**a–c**): $VE_{DIS}$ is exclusively mediated by a reduction in symptoms upon infection. Middle row (**d–f**): $VE_{DIS}$ is mediated by a combination of reduction in susceptibility to infection and reduction of symptoms upon infection. Bottom row (**g–i**): $VE_{DIS}$ is exclusively mediated by a reduction in susceptibility to infection. The columns correspond to assumptions that the single-dose efficacy (SDE) is low (left column, $VE_{DIS_1} = 18\%$), moderate (center column, $VE_{DIS_1} = 45\%$) or high (right column, $VE_{DIS_1} = 72\%$), corresponding 20%, 50%, or 80% of the full vaccine efficacy, $VE_{DIS} = 90\%$ assumed following two doses of vaccine, respectively. Shaded areas represent central 95% of 1000 simulations (uncertainty intervals, see SI for full details). Here, we assumed an effective reproductive number $R_{eff} = 1.1$.

(number of current SARS-CoV-2 active infections) at the beginning of vaccination in the optimal allocation strategies by considering 0.05 and 0.3% prevalence at the start of vaccination rollout. The optimal strategies were very similar irrespective of initial prevalence (Supplementary Fig. 13). Choosing the optimal strategy mattered most if vaccination started with lower prevalence: for example, with high SDE and 10% coverage, with 0.05% prevalence, the optimal strategy averted up to 26% more deaths than the high-risk strategy (optimal: 57% (95% UI: 47–64) vs. high-risk: 31% (95% UI: 29–33)), but it only averted a maximum of 14% more deaths (optimal: 44% (95% UI: 41–44) vs. high-risk: 30% (95% UI: 27–30)) if vaccination started with 0.3% prevalence. (Supplementary Fig. 14C, I).

**Sensitivity analysis for vaccination rate**. We next investigated the effect of vaccination rate in the optimal allocation strategies. Here, we assumed that vaccine was rolled out at 300 K doses per week (twice as fast as main scenario). At this rate, 100% of the population can be vaccinated with a single dose over the time span of six months. For low or high SDE, the optimal strategy was nearly identical to the one described in the main scenario. For moderate SDE, the optimal strategy prioritized the high-risk groups with two doses of vaccine (Supplementary Fig. 15). With enough vaccine to cover 50% of the population and administering 300K doses per week the optimal strategy averted ~ 12% more deaths compared to one distributing 150K doses per week. For example, with high SDE and 50% coverage, 87% (95% UI: 82–90)

and 74% (95% UI: 68–77) of deaths were averted compared with no vaccination, vaccinating at 300 and 150 K doses per week, respectively, Figs. S16C and 3C. Furthermore, at this rate and coverage, the optimal strategy to minimize deaths significantly mitigated transmission irrespective of the SDE, and temporary herd immunity was achieved if the vaccine had a high SDE (Supplementary Fig. 16g–i).

**Sensitivity analysis assuming a vaccine efficacy reducing infectiousness upon infection.** We identified the optimal allocation strategies assuming that a vaccine, in addition to all the effects previously described ($VE_{DIS} = 90\%$ after two doses with $VE_{SUS} = 70\%$ and $VE_{SYMP} = 66\%$) also reduced infectiousness upon infection by 70% ($VE_I = 70\%$) after two doses. For low and high SDE and for all coverage levels considered, the optimal strategies were very similar to the ones previously described. For moderate SDE, the optimal allocation was in fact the pro-rata strategy at low coverage (≤30%), and mixed vaccination strategies that included full coverage of the older adults with a one and two doses of vaccine (Supplementary Fig. 17) for higher coverage. All vaccination strategies averted slightly more deaths due to additional vaccine effects on infectiousness; this was more important at low coverage. With 10% coverage, the optimal strategies averted a maximum of 12% more deaths compared to the main scenario, regardless of the SDE (49% (95% UI: 41–54), 56% (95% UI: 47–61) and 66% (95% UI: 56–71) deaths averted for low, moderate and high SDE in this scenario versus 42% (95% UI: 35–45), 43% (95% UI: 36–47) and 53% (95% UI: 45–57) deaths averted for the main scenario, Fig. 3 and Supplementary Fig. 18, panels a–c. To note, the gains by using the optimal strategy were more evident in this scenario. For example, with low viral transmission ($R_{eff} = 1.3$), high SDE and 10% vaccination coverage, the optimal strategy averted up to 37% more deaths than the high-risk strategy (optimal: 68% (95% UI: 55–70) vs. high-risk: 31% (95% UI: 29–32) deaths averted) while the optimal strategy averted a maximum of 16% more deaths than the high-risk strategy in the main scenario), Fig. 3F and Supplementary Fig. 18F.

**Sensitivity analysis for the distribution of pre-existing immunity.** In this section we investigated the effect of different distributions of pre-existing immunity on the optimal allocation strategies. To this end, we considered two additional age distributions of pre-existing immunity: one using the reported distribution of the number of cases in WA state[24] as of March 2021 and a second one using the reported distribution of the number of cases in two Indian states[36] as of July 2020. For each of these distributions, we repeated the optimization routine (assuming low viral transmission and 150 K doses administered weekly). It is important to note that in both cases, these distributions are potentially over-representing age groups that are more prone to disease and under-representing those for whom SARS-CoV-2 infections result in a milder or asymptomatic disease. The optimal allocation strategies for both distributions resulted in nearly identical allocations to those presented in the main section, with minor differences observed for 20–30% coverage and a moderate SDE (Fig. 1 and Supplementary Figs. 19 and 20.)

**Sensitivity analysis assuming increased transmission due to establishment of new variants.** We repeated our main analysis assuming a baseline $R_0 = 4$, representing an overall increase in transmission due to the circulation of new more transmissible variants in the population. We assumed that vaccines were equally effective against these new variants. Here, the four levels of social distancing interventions considered (assuming 20% pre-existing immunity) resulted in effective reproductive numbers of

$R_{eff} = 1.5$, 1.7, 2.0, and 3.2. The optimal allocation strategies under this scenario were, as expected, identical to those given in the main text when we assumed $R_0 = 3$ and a high level of viral transmission (Fig. 5 and Supplementary Fig. 21). In line with our previous results, under increased baseline transmission, the optimal strategy averted roughly as many deaths as the high-risk strategy for almost all levels of coverage with a maximum of 11% more deaths averted (high SDE, $R_{eff} = 1.5$, 10% coverage, optimal: 41% (95% UI: 40–42) vs. high-risk: 30%, (95% UI: 29–31)), and averted a maximum of 46% more deaths than the pro-rata strategy (low SDE, $R_{eff} = 1.5$, 30% coverage, optimal: 58% (95% UI: 47–62) vs. pro-rata:12% (95% UI: 10–14)), Supplementary Fig. 22.

**Sensitivity analysis assuming a lower vaccine efficacy.** We next investigated the effect of vaccine efficacy in our optimal strategies. In this section, we considered a vaccine with a lower vaccine efficacy, so that $VE_{DIS} = 72\%$ after two doses (similar to that reported in[37]) with $VE_{SUS} = 49\%$ and $VE_{SYMP} = 46\%$. Our results were not sensitive to this assumption. The optimal allocation strategy was almost identical to that presented in the main text, with minor differences for moderate SDE and medium coverage (Fig. 1 and Supplementary Fig. 23). As expected, a lower vaccine efficacy resulted in a lower number of deaths averted (maximum deaths averted: $R_{eff} = 1.1$, 50% coverage, high SDE, optimal: 65% (95% UI: 59–68) vs. 74% (95% UI: 68–77)), Fig. 3 and Supplementary Fig. 24.

## Discussion

COVID-19 vaccination has begun in several countries, and more countries will start in the upcoming months. As demand will far exceed supply in the initial months of vaccine deployment, vaccine doses will need to be prioritized. The current strategies of most countries consider vaccination with full dosage (two doses), but some have proposed vaccinating twice as many people with a single dose and delaying the second dose[38]. An intense debate about how best to use the available vaccine is ongoing[15,39,40]. Here, we show that there is no universal answer to this question. Pairing a mathematical model parameterized using the evidence to-date on the efficacy of COVID-19 vaccines with optimization algorithms, we explore the use of single-dose campaigns and mixed vaccination campaigns, with some people receiving one dose and others receiving two doses, and we find that the optimal use of resources depends primarily on the level of single-dose efficacy, in agreement with ref. [17]. If a single dose of vaccine is highly efficacious and introduced under stringent social distancing interventions with low viral transmission, our results suggest that campaigns that optimally distribute a single vaccine dose to more people are more effective at averting deaths than a two-dose vaccination campaign prioritizing subpopulations at high risk of COVID-19 severe disease and death. Previous work for other infectious diseases[41–43] has reached similar conclusions. Furthermore, these results show that vaccinating with a single dose at a faster rate could result in temporary herd immunity, in agreement with previous work[44]. Importantly, as more vaccine becomes available, additional vaccination campaigns may be used to cover everyone with the full two doses of vaccine to provide full protection. In contrast, however, when SARS-CoV-2 transmission is moderate or high ($R_{eff} = 1.5$ or 2.4 in our model), we find that a two-dose campaign from the outset is optimal.

In addition, we show that optimal distribution of available vaccine doses across subpopulations depends strongly on the level of transmission. If the ongoing transmission in the community is well-controlled with stringent non-pharmaceutical interventions in place, the optimal strategy allocates vaccine to both the high-

risk and the high-transmission groups, consistent with previous work[45]. In contrast, if the level of transmission is moderate or high, it is optimal to directly protect those at high risk of severe disease and death, also in agreement with previous results[20,46]. Our results highlight the absolute necessity of maintaining social distancing throughout vaccination[47–49]: if social distancing interventions are lax before vaccination is advanced, or if vaccination is not rolled out fast enough, then the current epidemic wave will be over long before vaccination campaigns are completed and the effect of vaccination will be limited.

While high vaccine efficacy against COVID-19 has been reported for vaccines requiring two doses, both for those currently administered in the US (Pfizer and Moderna) and for two others administered in other parts of the world (Sputnik V and AstraZeneca), other effects of COVID-19 vaccines require further evaluation, including their effects on preventing SARS-CoV-2 infection and on infectiousness. To account for these gaps in knowledge, we investigated the optimal vaccine allocation under three possible vaccine profiles consistent with the observed vaccine efficacy against disease, and we found that the optimal vaccination strategy depends on the profile. Our analysis showed that a vaccine that mostly mitigates symptoms but does not reduce the risk of infection should be prioritized to the oldest age groups at full dosage. In contrast, the optimal strategy for a vaccine that provides at least moderate protection against infection includes more balanced dose distribution across age groups with larger proportions assigned to one-dose vaccinations. Similar to ref. [49], we found that a vaccine that only prevents disease upon infection will have limited population impact, whereas a vaccine preventing infection will have a larger effect in reducing population transmission and subsequent morbidity and mortality. These results underscore the need for thorough studies to evaluate all of the vaccine effects.

Beyond the impact on infection and disease burden, there are additional arguments for considering single-dose vaccination, including greater equity achieved in distributing a scarce commodity (vaccine)[15], reduced reactogenicity following the first versus the second dose of the mRNA vaccines[3,18], and the potential for greater population uptake and adherence to a single-dose regimen. However, strictly following the two-dose vaccination schedule might also be important, as the efficacy of a two-dose regimen with a longer interval between doses has not been evaluated systematically in efficacy trials, especially in the context of emerging variants[50]. Policy-makers would ideally consider these issues in evaluating possible vaccination strategies.

Here, we report the optimal use of resources as determined by mathematical optimization. In practice, other factors need to be considered when allocating vaccine. These include differences in differences in potential behavioral effects associated with vaccination (e.g., behavioral disinhibition), perceived risk and vaccine hesitancy amongst different age groups, ethical and logistical considerations. We minimized five different metrics of disease burden, and our results suggest that the optimal use of vaccine depends on the metric chosen. These metrics were chosen based on the prevailing scientific discussions surrounding COVID-19 vaccine prioritization[51,52]. In particular, we chose to optimize cumulative number of deaths as opposed to years of life lost (YLL), and this resulted in attributing more weight to vaccinating the oldest age groups, which are at greatest risk of death. Other metrics like YLL would give rise to different optimal strategies, that may favor younger age groups. Furthermore, new evidence is emerging about the long-term consequences of COVID-19 disease. This might be an important factor to incorporate in the metrics of disease burden in future work.

We quantified the advantages and disadvantages of two vaccination prioritization schedules that closely mimic current

guidelines—pro-rata vaccination and vaccination of groups at high risk of disease—and identified when either of these coincides with our computed optimal allocation strategy, or achieves similar public health impact. While some of the optimal allocation strategies may be difficult to implement, our results can be used to guide the development of mixed vaccination strategies, where some subpopulations receive one dose and others receive two doses, thereby achieving a balance between rapid coverage and full protection of those most at risk of severe disease and death.

Our work has several limitations. Our model assumed that asymptomatic and symptomatic infections confer equal protection, but asymptomatic infections could result in weaker protection[53]. We assumed that naturally and vaccine-induced immunity will be at least six months, but the duration of immunity is not yet known; durable immune responses to the Moderna mRNA-1273 vaccine have been found through 3 months post-second-dose[54]. Ongoing phase 3 trials will establish the durability of vaccine efficacy, with participants followed 1 to 2 years post-last vaccination. If immunity is short-lived, then our results are valid only for that time frame. We also assumed that vaccinating previously infected individuals would have no effect on their immunity. However, limited data emerging suggests that previous infections might "prime" the immune response, with only one dose boosting immunity[55]. This would be an important consideration for future work when evaluating vaccine allocation over longer time spans with waning immunity. We use age-stratified hospitalization rates based on data from Wuhan, China[56] and mortality rates based on data from France[57]. These rates strongly depend on comorbidities (e.g., heart disease, diabetes, etc.) that are country-dependent. It is then important to determine country-based estimates of these rates to adequately parameterize models. Deterministic models implicitly result in exponential durations for each disease state, and they can overestimate infection dynamics. While comparisons with and without vaccination strategies would not be affected by this, it is possible that our peak hospitalizations are overestimated and are occurring at a faster pace than in reality. Further, for mathematical and computational tractability, we used a model that does not account for geographic movement or complex contact patterns and age was our sole risk factor. In reality, other factors, such as occupation, have been linked to increased risk of acquisition and severe disease[58,59]. Because of systemic social inequalities, several studies have shown that in certain countries people from racial and minority groups are at increased risk of infection and death from COVID-19[60]. Social distancing interventions are being constantly changed and adapted to new challenges when transmission is high, but we kept our social distancing interventions fixed for the duration of the simulations. We included children as possible recipients of vaccine in our analysis, but vaccines are not currently licensed for individuals under 16 years old. However, vaccine trials for younger children are underway for the AstraZeneca and Pfizer vaccines[61] and other vaccine studies in children are being planned, so it is possible that one or more COVID-19 vaccines will be licensed in children within the next 6 months. Importantly, in our model contact rates of children were greatly reduced (by 90% in the contact rates of children at school) and, while the algorithm can in principle select strategies vaccinating children, it seldom does so. In order to run the optimization, we considered a fixed implementation scheme to model vaccination campaigns. We chose this particular implementation because it most closely mimics current vaccine campaigns around the world and it would favor minimizing deaths, but it is important to note that this implementation scheme would favor the high-risk strategy and would undermine the pro-rata strategy. In that sense, our results regarding the pro-rata strategy are conservative. Moreover, we used the same

implementation for all the disease metrics considered, but other implementations would be more appropriate for different disease metrics. Finally, we have determined optimal allocation strategies in the context of considerable uncertainty as to COVID-19 vaccine profiles and vaccine rollout; once vaccine profile, vaccination rates and coverages for specific countries are known, we welcome validation with more complex models.

There are reports of new and more transmissible SARS-CoV-2 variants first identified in the UK, South Africa, and Brazil that are spreading rapidly and circulating in several parts of the globe. It is still unknown how efficacious currently available vaccines will be against each of these variants, with some vaccines exhibiting a decreased efficacy against some of them[29,62–65]. Our results show that if the goal is to minimize transmission as a way to minimize the spread of these new variants, then it is optimal to vaccinate the high-transmission groups with one or two doses depending on the SDE. However, a potential concern with single-dose vaccination is that vaccinating large numbers of people with a regimen with suboptimal efficacy may allow selection to drive the emergence of new vaccine-resistant variants that can rise rapidly in frequency[66].

While emerging data suggest that a single dose of the three COVID-19 vaccines (that require two doses) with regulatory licensure or approval in the US and UK might confer high efficacy[2,37,67], the duration of protection remains unknown. Other vaccines that require two doses, such as the oral cholera vaccine, are highly effective after a single dose but the protection provided is short-lived compared to that provided by the full two-dose regimen[68]. If single-dose immunity lasts for at least 6 months, our results show that if viral transmission is low, single-dose vaccination campaigns, which are much easier to implement, are the optimal use of resources in the short term, with the goal to fully vaccinate the entire population in the long term. In the absence of phase 3 efficacy data on single-dose vaccination, it will be crucial for vaccine safety systems to capture any breakthrough infections that occur among individuals receiving vaccination under population campaigns—especially among those receiving a single dose; and for longitudinal immune responses to be measured in clinical trial participants who received only one dose. Our work suggests that it is an absolute imperative to quickly and fully determine the peak and duration of efficacy of single-dose vaccinations, as these data are needed to support further investigation of mixed vaccination campaigns, which have great potential to more quickly contain the pandemic.

## Methods

**Transmission model**. We built upon our previous model of SARS-CoV-2 transmission and vaccination with 16 age groups: 0–4, 5–9, 10–14, 15–19, 20–24, 25–29, 30–34, 35–39, 40–44, 45–49, 50–54, 55–59, 60–64, 65–69, 70–74, and 75+[20]. We used the population of Washington state (7.6 million people[69]) and US demographics[70]. For each age group $i$, our model tracks susceptible $S_i$, exposed $E_i$, asymptomatic $A_i$, pre-symptomatic $P_i$ and symptomatic infected individuals classed by disease severity (assumed to be equally infectious). Symptomatic individuals have one of three fates: they become mildly symptomatic $I_i$, hospitalized in a non-ICU ward $H_i$, or hospitalized requiring intensive care, $ICU_i$. After infection, individuals move to the respective recovered classes: recovered asymptomatic, mildly symptomatic, non-ICU hospitalized and ICU hospitalized (denoted by $RA_i$, $R_i$, $RH_i$, and $RC_i$, respectively). Individuals may receive one or two doses of vaccine (analogous compartments indexed by $j = 1, 2$, respectively) (Supplementary Fig. 1 and Vaccination section below). We assumed that asymptomatic are less infectious than symptomatic (not hospitalized) infections but confer equal immunity. Further, we assumed that both naturally induced and vaccine-induced immunity are long lasting, so that there is no waning during the time period analyzed.

We used a previously published age-structured contact matrix $\mathcal{M}$ for contact patterns in the US[21] and corrected for reciprocity. We assumed a baseline $R_0 = 3$ in the absence of any social distancing interventions. We also considered an alternative scenario resulting in increased transmission due to the co-circulation of the original and new variants that would result in a baseline $R_0 = 4$. To simulate social distancing interventions, we modified the matrices given in[21] (matrices

corresponding to contacts at "home", "work", "other locations", and "school") according to Supplementary Table 3 and Supplementary Fig. 2 to obtain an effective reproduction number $R_{\text{eff}} = 1.1, 1.3, 1.5$, or $2.4$ (assuming 20% pre-existing immunity). We informed these parameter choices by the different social distancing interventions that have been enacted in WA state over the course of the pandemic and by mobility data[71–76]. We used estimated age-specific proportions of infections that require hospitalization and critical care from[56]. Estimated proportions of hospitalizations that result in death were taken from ref.[57]. These reports have different age brackets than those in our model, so we combined the brackets according to the proportion of the population in each bracket (e.g., for the oldest group in our model, ≥75, we weighted the rates in ref.[56] by the relative percentages of the US population aged 75–80 and ≥80). Hospitalized individuals are assumed no longer infectious. Number of hospital beds (ICU and general beds) available in WA state as well as state goals were taken from ref.[26]. Supplementary Table 2 summarizes the parameter values, ranges, and sources for the model.

Simulations were run with initial conditions set to a 20% of the population with pre-existing immunity, distributed proportionally to population size (pro-rata) and disease severity, respectively (additional scenarios, 10% pre-existing immunity, 20% pre-existing immunity with different age distributions, see Sensitivity Analysis). In addition, simulations were run with 7615 infections (equivalent to 0.1% of the total population) initially, distributed among the infectious symptomatic and asymptomatic infectious compartments ($A_{ij}$, $A_{V,ij}$, $I_{ij}$, $I_{V,ij}$, $H_{ij}$, $H_{V,ij}$, $C_{ij}$, and $C_{V,ij}$). We ran additional scenarios with an initial prevalence of 0.05% and 0.3% of the total population.

**Vaccination**. Following the ideas of Halloran et al.[28], we assumed a leaky vaccine (that is, a vaccine that confers partial protection to all the vaccinees) that can have three effects on the vaccinated individuals. First, the vaccine can reduce the probability of acquiring a SARS-CoV-2 infection, (we denote this effect by $VE_{\text{SUS}}$). Second, the vaccine can also potentially reduce the probability of developing COVID-19 symptoms upon infection (referred to as $VE_{\text{SYMP}}$), or third, reduce the infectiousness of vaccinated individuals (referred to as $VE_I$), Supplementary Fig. 3B. There is a multiplicative relationship between the vaccine efficacy against laboratory-confirmed COVID-19 disease, $VE_{\text{DIS}}$, $VE_{\text{SUS}}$, and $VE_{\text{SYMP}}$[77], so that

$$VE_{\text{DIS}} = 1 - (1 - VE_{\text{SUS}})(1 - VE_{\text{SYMP}}). \tag{1}$$

A vaccine highly efficacious against disease could be either mediated mainly by protecting vaccinated individuals against infection (high $VE_{\text{SUS}}$), or mainly by preventing them from developing symptoms once infected (high $VE_{\text{SYMP}}$), or a combination of both. A vaccine with a high $VE_{\text{SUS}}$ or a high $VE_I$ (irrespective of $VE_{\text{SYMP}}$) would have a bigger effect on the transmission dynamics of SARS-CoV-2, resulting in a greater population impact than one mediated primarily by $VE_{\text{SYMP}}$. In fact, a vaccine mediated exclusively by $VE_{\text{SYMP}}$ might have only a direct effect, protecting only those vaccinated (this would be the case if asymptomatic and symptomatic individuals are equally infectious). We denote by $VE_{\text{SUS}_1}$, $VE_{\text{SYMP}_1}$, and $VE_{I_1}$ the single-dose vaccine efficacies for susceptibility, symptomatic infection upon infection, and infectiousness upon infection respectively. Values for all the combinations of vaccine efficacy profiles considered can be found in Supplementary Table 1.

The equations for this model are given by

Unvaccinated:

$$\frac{dS_i}{dt} = -m_i \lambda\, S_i,$$

$$\frac{dE_i}{dt} = m_i \lambda\, S_i - \gamma_E E_i,$$

$$\frac{dA_i}{dt} = (1 - k_i)\gamma_E E_i - \gamma_A A_i,$$

$$\frac{dP_i}{dt} = k_i \gamma_E E_i - \gamma_P P_i,$$

$$\frac{dI_i}{dt} = \gamma_P P_i - (1 - h)\gamma_I I_i - h(1 - c)\sigma\, I_i - hc\sigma\, I_i,$$

$$\frac{dH_i}{dt} = h(1 - c)\sigma\, I_i - \gamma_H H_i,$$

$$\frac{dICU_i}{dt} = hc\sigma\, I_i - \gamma_C ICU_i,$$

$$\frac{dRA_i}{dt} = \gamma_A A_i,$$

$$\frac{dR_i}{dt} = \gamma_I I_i,$$

$$\frac{dRH_i}{dt} = \gamma_H H_i,$$

$$\frac{dRC_i}{dt} = \gamma_C ICU_i, \tag{2}$$

Vaccinated j=1,2 denotes vaccination with one or two doses, respectively):

$$\frac{dS_{ij}}{dt} = -(1 - VE_{SUS_j})m_i\lambda\ S_{ij}\ ,$$

$$\frac{dE_{ij}}{dt} = (1 - VE_{SUS_j})m_i\lambda\ S_{ij} - \gamma_E E_{ij}\ ,$$

$$\frac{dA_{ij}}{dt} = (1 - k_i(1 - VE_{SYMP_j}))\gamma_E E_{ij} - \gamma_A A_{ij}\ ,$$

$$\frac{dP_{ij}}{dt} = k_i(1 - VE_{SYMP_j})\gamma_E E_{ij} - \gamma_P P_{ij}\ ,$$

$$\frac{dI_{ij}}{dt} = \gamma_P P_{ij} - (1 - h)\gamma_I I_{ij} - h(1 - c)\sigma\ I_{ij} - hc\sigma\ I_{ij}\ ,$$

$$\frac{dH_{ij}}{dt} = h(1 - c)\sigma\ I_{ij} - \gamma_H H_{ij}\ , \tag{3}$$

$$\frac{dICU_{ij}}{dt} = hc\sigma\ I_{ij} - \gamma_C ICU_{ij}\ ,$$

$$\frac{dRA_{ij}}{dt} = \gamma_A A_{ij}\ ,$$

$$\frac{dR_{ij}}{dt} = \gamma_I I_{ij}\ ,$$

$$\frac{dRH_{ij}}{dt} = \gamma_H H_{ij}\ ,$$

$$\frac{dRC_{ij}}{dt} = \gamma_C ICU_{ij},$$

where $\lambda$ is the force of infection given by

$$\lambda = \beta \sum_{k=1}^{16} \frac{\mathcal{M}}{N_k}[r_A(A_k + \psi_1 A_{k1} + \psi_2 A_{k2}) + r_P(P_k + \psi_1 P_{k1} + \psi_2 P_{k2}) \\ + (I_k + \psi_1 I_{k1} + \psi_2 I_{k2})], \tag{4}$$

with $\psi_1 = 1 - VE_{I_1}$, $\psi_2 = 1 - VE_{I_2}$, and $\mathcal{M}$ is the sum of the contact matrices given in ref. [21], corrected for reciprocity and weighted by the multipliers given in Supplementary Table 3.

**Modeling vaccination campaigns**. Vaccination campaigns were modeled assuming 150,000 doses of vaccines delivered weekly, over the span of ~ 6 months (28 weeks). At this rate, 50% of the population can be vaccinated over this time period with a single dose (25% with two doses). We also analyzed an alternative scenario with 300,000 doses delivered weekly (corresponding to vaccinating 100% of the population with one dose over the same time period).

For each vaccination coverage and strategy considered, we computed within each age-group the fraction of susceptible individuals among all those individuals in that group who could have sought the vaccine (susceptible, exposed, infected pre-symptomatic, infected asymptomatic, and recovered asymptomatic populations), and used that fraction as the fraction of people who were actually vaccinated in each age-group, while assuming that the remaining vaccine would be wasted. As it is expected that vaccine supplies will ramp up considerably over the second half of 2021 and into 2022, we focused on the first few months of vaccine availability and set 6 months as our time horizon, both for the optimization and for the population impact.

We implemented the vaccination campaigns following current implementation in the US, that is, vaccinating individuals starting always with the oldest age-group and moving sequentially in decreasing order across the vaccine groups. For vaccination strategies that included vaccination with one and two doses of vaccine, we vaccinated first all the age-groups receiving two doses and then those receiving one dose. For example, if a particular strategy allocates vaccine as follows: 10% of adults aged 20–50 vaccinated with one dose, 10% of adults aged 50–65 vaccinated with two doses, 15% of adults aged 65–75 vaccinated with one dose and 40% of adults aged 75 and older vaccinated with two doses, then the vaccination in our model goes as follows: (1) vaccinate 40% of adults aged 75 and older with one dose for as many weeks as necessary, then repeat this to vaccinate them with their second dose. (2) Then we will vaccinate 10% of the adults aged 50–65 with two doses (similarly to the previous steps, in two rounds). (3) Then we will vaccinate 15% of adults aged 65–75 with one dose, and finally (4) we will vaccinate 10% of those aged 20–50 with a single dose.

We chose this implementation because it closely reflects the current implementations around the world, with interest to minimize death. However, it is important to bear in mind the potential artifacts of this implementation. First, this implementation tends to favor the high-risk strategy but it might disadvantage other strategies, in particular the pro-rata strategy. Second, this implementation works well to minimize deaths but might not ideal to minimize infections, since the groups driving the epidemic (adults aged 20–49 and children) are the last to be vaccinated. Furthermore, due to this implementation, a particular allocation can, in principle, avert fewer infections or deaths than do allocations with less coverage. This is because in our model, as more vaccine is available, it is always distributed to older adults first. More coverage implies more time spent vaccinating the older adults vaccination groups, who transmit the least, hence delaying vaccination to the younger age groups, who are those transmitting most. This was apparent for the pro-rata strategy when minimizing infections.

**Optimization**

*Objective functions.* We performed the optimization routine to minimize five different objective functions: cumulative number of infections, cumulative number of symptomatic infections, cumulative number of deaths, maximum number of hospitalizations not requiring intensive care, and maximum number of hospitalizations requiring intensive care. For each of these, we ran the deterministic model for 6 months (our time horizon).

*Optimization.* Here, we describe our optimization routine, adapted from our previous work[20]. We randomly selected 10,000 points on a coarse grid[78] of the unit simplex in the vaccination group space (the set of vectors $(v_1, v_2, ..., v_5)$ with non-negative entries such that $\sum_{i=1}^5 v_i = 1$). The grid was chosen so that the unit simplex was divided into 0.05 units and was computed in Sage[79]. For each point in the coarse grid, the five objective functions were evaluated. For each of these objective functions, we selected the best 25 decision variables obtained in the grid search, the pro-rata allocation vector, the high-risk allocation vector and an additional 25 decision variables sampled uniformly from the unit simplex[80], and used these 52 points as initial points for the Nelder-Mead minimizer implemented in SciPy[81,82]. Full details of the optimization routine can be found in ref. [20].

The optimization for each combination of parameters (vaccination coverage, values of different vaccine profiles, level of viral transmission, etc.) was run independently. This, together with the way vaccination campaigns was implemented (for the reasons explained above), can result in the optimal allocation strategies that are not always monotonic functions of coverage.

We used a heuristic method because it is fast, and using a method that returns a guarantee of optimality is very difficult for complex black-box objective functions. We took a very large sample of the entire feasible set to ensure a thorough search to increase the chance that our solution is nearly globally optimal. Furthermore, the sensitivity analysis shows that the optimal solutions we found are robust to changes in parameters and, even though we cannot guarantee they are globally optimal, our solutions either coincide with or outperform the pro-rata and high-risk strategies, so they are relatively optimal in that sense.

**Uncertainty analysis**. We examined the uncertainty in the output measures (percentage of infections averted and percentage of deaths averted, etc.) arising from uncertainty surrounding the model parameters. The following model parameters were varied for this analysis: the duration of the latent period, the duration of the pre-symptomatic, symptomatic and asymptomatic infectious periods, the relative infectiousness of the pre-symptomatic infected individuals, and the proportion of infections that are asymptomatic. We sampled parameter sets from predetermined distributions as follows: for the duration of the latent period, the duration of the pre-symptomatic, symptomatic, and asymptomatic infectious periods we sampled gamma distributions with means given in Supplementary Table 2. For the proportion of asymptomatic infections and the relative infectiousness of the pre-symptomatic individuals we used truncated normal distributions with means and ranges given in Supplementary Table 2. We then sampled 1000 parameter sets and evaluated all three strategies (optimal, pro-rata, and high-risk) for each of those sets. We also ran the simulation in absence of vaccination. We then computed the outcomes of interest and removed the the top and bottom 2.5%. The shaded areas presented in the figures are the result of this analysis.

**Reporting summary**. Further information on research design is available in the Nature Research Reporting Summary linked to this article.

## Data availability

The data used in this paper are fully described in the methods and references. Sources for all model parameters are provided in Supplementary Tables 1, 2, and 3.

## Code availability

Code available at: https://github.com/lulelita/one_vs_two_doses

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

## Acknowledgements

We thank Peter Gilbert, Dimitri Leemans, and Natalia García-Colín for helpful discussions. L.M. was supported by Fred Hutchinson Cancer Research Center discretionary funds awarded to Peter Gilbert. The Scientific Computing Infrastructure at Fred Hutch was supported by ORIP grant S10OD028685. H.J. was supported by R56AI143418 and R01CA152089 from the National Institutes of Health. D.D., J.T.S., and D.A.S. were supported by NU38OT000297-02 from Centers for Disease Control and Prevention through a cooperative agreement with the Council of State and Territorial Epidemiologists. L.M., H.J., and D.D. were supported by UM1AI068635 from the National Institutes of Health.

## Author contributions

L.M. conceived the study, conducted the analysis, and wrote the first draft of the manuscript. J.E. assisted with the optimization, analysis, figures. T.L., D.D., J.T.D., D.A.S., and H.J. contributed to writing the manuscript.

## Competing interests

The authors declare no competing interests.
