## [Peer Review File · Nature Communications]

REVIEWER COMMENTS

Reviewer #1 (Remarks to the Author):

General comments:

This is a well-written manuscript on the highly relevant topic of choosing how to roll-out vaccination campaigns when vaccine supply is limited: one or two doses per person? The authors optimise the design of vaccine campaigns based on relevant metrics for infections, death, and healthcare burden, although not morbidity (e.g. long-term consequences, although these may be derived from cumulative number of infections by age). The authors are to be commended for compactly visualising the dizzying (but relevant) number of scenarios. I encourage the authors to put the results even more in a public health context by discussing the impact of changes in population behaviour during vaccination roll-out, age-dependent uptake of actual vaccination, and the impact of the types of optimisation targets that were chosen (details below).

Results:

Regarding: "We built upon our previous model of SARS-CoV-2 transmission and vaccination [20]. Briefly, we developed an age-structured mathematical model with the population of Washington state." Please clarify what kind of mathematical model is employed (deterministic, stochastic ODE, individual-based, ...?).

The authors assume that 20% or 10% of the population has pre-existing exposure (immunity) to Covid-19, which is probably relevant to the Washington State setting. However, the distribution of immunity over age is not mentioned, although this should be expected to be a critical determinant of what the best optimal vaccination campaign design is. To make the results of this study more widely applicable, I urge the authors to consider the age distribution of pre-existing immunity (which may very well vary between countries, and perhaps LMIC and first world settings) as a variable for the scenario analyses.

Are the assumed transmission scenarios (R_t) conditional on the pre-existing prevalence of immunity, the prevalence of infectious cases, and the relative infectiousness of asymptomatic cases? This would mean that the different transmission scenarios represent very different settings in terms of contact rates; it would be good if the author could show (in supplements) how plausibly different these scenarios are in terms of contact rates. Or alternatively (and better I think), define transmission scenario based on a set of three contact rates that correspond to some values of R_t for one of the scenarios for pre-existing immunity, prevalence of infectious cases. As the relative infectiousness of asymptomatic individuals is a biological parameter (i.e. not context-related), this

set of three contact rates would have to be redefined for the alternative assumptions of infectiousness of asymptomatics (again corresponding to the same values of R_t).

Figure 2: It would be very helpful for the reader to be able to also see the predictions for other calibration metrics (deaths, healthcare burden), even though the campaign designs were optimised to reduce number of infections. This could be done with extra rows of panels, or as supplement to which can be referred in the figure caption.

Discussion:

The values of R_t mentioned for moderate and high transmission don't match the range of values reported in the results section.

Please discuss more about the potential impact of changes in risk behaviour in the population as vaccination is rolled out due to perceived declining risk. Sure, maintaining social distancing is paramount. Will people do it, though? Probably not as well as we would like, especially the "high transmission" groups. So, based on the findings, which of the examined strategies should be expected to be the most robust to lax social distancing? Please also discuss the impact of potentially lower willingness to take up vaccination among younger age groups, and in particular, the high-transmission age group.

Please highlight or discuss the consequence of using "deaths" as an optimisation metrics instead of "years of life lost" (which would favour preventing deaths in younger individuals).

Please highlight or discuss the relevance of long-term consequences of Covid-19, and how this may be derived from this work and how that might change optimal designs.

Review of NCOMMS-21-08478

Optimizing vaccine allocation for COVID-19 vaccines: potential role of single dose vaccine.

Comments:

This paper tackles a very important and relevant problem playing out literally today in terms of what is the role for a single dose vaccine regimen versus a two-dose regimen and can simulation-based modeling reveal optimal allocation strategies that can help guide actual practice on the ground. The research team conducted a very comprehensive set of analyses taking into account many potential factors using a deterministic modeling framework. They then conducted an extensive sensitivity analysis whereby certain assumptions were varied and the results compared with respect to five important metrics of disease burden. Models are constructed using Washington state as a framework for inferences. Important conclusions from the simulations are reached and an extensive discussion focusing very much on study limitations is given (which is to be very much lauded). In fact lots of room is dedicated to fleshing out these limitations and how things could be improved.

The modeling descriptions in the appendix look to be complete and the simulations are summarized nicely using both graphical and numerical approaches. Overall, this is a very nice contribution to the research literature. I do have a few specific comments which would improve the paper further:

1. Since there are multiple variants circulating in the US now, it seems to me that rather than varying a single R_{eff} as a tuning parameter, that a mixture of R_{eff} s should be used each reflecting a different variant.
2. Other public health measures are still be encouraged and are being adapted to varying degrees by state or local areas. Building this kind of adoption into the modeling framework can be useful since the vaccine program efficacy will also interact with these other public health measures like masking and social distancing.
3. Multiple vaccines are being used at the same time – some with differing efficacies at preventing infection. This should be embedded into the modeling structure.
4. Even though the focus is in the Washington state area, the patchwork process by which the US has rolled out programs could also be simulated. This would mean allowing for more local structure in the modeling.
5. Everything is done using deterministic models. Some small discussion around what might be expected if stochastic variations of the models were used instead.

Reviewer #3 (Remarks to the Author):

This is an interesting manuscript dealing with the important issue of optimizing vaccine allocation.

I have some minor comments the authors might like to address:

p2: I am surprised you have not included your own paper from 2015 perhaps being the first to address one vs two vaccine responses.

p4: I did not check the S.M: but I strongly encourage that you explicitly give the contact matrix between different age groups

p6: Why do you end at 50% receiving one dose? The question remains interesting even when having vaccines for e.g. 120%

p7: Do you really know it is optimal (mathematically)? Perhaps a few words about the algorithm in main text or SM.

p7: How is parameter uncertainty expressed? Give specific reference to SM.

p7, Results: I am surprised that the optimal strategy often vaccinates in all age groups. I would have expected filling one age group with at least one dose before going to another the next age group. Is the explanation to this that much of the contact mixing is on the diagonal? Please comment more in text. Also, why is the optimal fractions more or less equal between age groups when SDE is high? Comment.

Discussion: policies which recommend to vaccine fractions in all age-groups are most likely hard to implement practically. Comment.

p17, mid-paragraph. There are of course also arguments for going for two vaccines. Even if these are more obvious I would state them, e.g.: vaccine guarantee, a second dose much later may not have same overall effect.

p18: a deterministic system defined by rates implicitly imply exponential durations in states. Don't change model, but acknowledge this.

p18, regarding individual with anti-bodies. To me it is obvious that when having limited vaccine supply these individuals should not be vaccinated at all. Did you address this?

p19, line f4: peak hospitalizations are overestimated and happen too early.

p19, middle. I don't at all understand comment that you only consider a fixed prioritization scheme. I thought the whole idea was that you compared different orders?

We thank the reviewers for their very helpful comments. Please see our answers below in blue.

Reviewer #1 (Remarks to the Author):

General comments:

This is a well-written manuscript on the highly relevant topic of choosing how to roll-out vaccination campaigns when vaccine supply is limited: one or two doses per person? The authors optimise the design of vaccine campaigns based on relevant metrics for infections, death, and healthcare burden, although not morbidity (e.g. long-term consequences, although these may be derived from cumulative number of infections by age). The authors are to be commended for compactly visualising the dizzying (but relevant) number of scenarios. I encourage the authors to put the results even more in a public health context by discussing the impact of changes in population behaviour during vaccination roll-out, age-dependent uptake of actual vaccination, and the impact of the types of optimisation targets that were chosen (details below).

Results:

Regarding: "We built upon our previous model of SARS-CoV-2 transmission and vaccination [20]. Briefly, we developed an age-structured mathematical model with the population of Washington state." Please clarify what kind of mathematical model is employed (deterministic, stochastic ODE, individual-based, ...?).

We have now added the type of model to that sentence, and it reads:

"Briefly, we developed a deterministic age-structured mathematical model with the population of Washington state (7.6 million people) and US demographics divided into 16 age groups with contact rates given in [21], adjusted for reciprocity."

The authors assume that 20% or 10% of the population has pre-existing exposure (immunity) to Covid-19, which is probably relevant to the Washington State setting. However, the distribution of immunity over age is not mentioned, although this should be expected to be a critical determinant of what the best optimal vaccination campaign design is. To make the results of this study more widely applicable, I urge the authors to consider the age distribution of pre-existing immunity (which may very well vary between countries, and perhaps LMIC and first world settings) as a variable for the scenario analyses.

We have added a line in the main text explaining how the pre-existing immunity was calculated and added two scenarios where we took a different distribution of immunity: one that mimics the distribution of cases in WA state as of March 2021 and one that mimics the distribution of cases in two Indian states as of July 2020 (see within Sensitivity Analysis, subsection entitled "Results assuming a different distribution of pre-existing immunity").

In addition, to make the paper more accessible to a wider audience, we created a Lay summary figure, pointing out the vaccine allocation strategies as functions of the background transmission and the single dose efficacy (fig. 8) and added a sentence in the results pointing to this figure:

"A summary of the optimal allocation strategies for different combinations of SDE and background transmission for minimizing total infections, hospitalizations and deaths is given in Fig. 8."

Are the assumed transmission scenarios (R_t) conditional on the pre-existing prevalence of immunity, the prevalence of infectious cases, and the relative infectiousness of asymptomatic cases? This would mean that the different transmission scenarios represent very different settings in terms of contact rates; it would be good if the author could show (in supplements) how plausibly different these scenarios are in terms of contact rates. Or alternatively (and better I think), define transmission

scenario based on a set of three contact rates that correspond to some values of R_t for one of the scenarios for pre-existing immunity, prevalence of infectious cases. As the relative infectiousness of asymptomatic individuals is a biological parameter (i.e. not context-related), this set of three contact rates would have to be redefined for the alternative assumptions of infectiousness of asymptomatics (again corresponding to the same values of R_t).

Thank you for this comment. We realized that the assumptions were not clearly stated. In our model, the assumed transmission scenarios are the result of the following parameters: a baseline R_0 , (in the main text, we assumed $R_0 = 3$, the level of pre-existing immunity in the population, and the social distancing interventions being adopted. The relative infectiousness of the asymptomatic individuals also plays a role in R_{eff} , but we explored the effect of this parameter on our results separately in the sensitivity analysis section. We have now changed the wording in the results section to make this clear, it reads now as follows:

“We considered a baseline $R_0 = 3$ (alternative scenario $R_0 = 4$, Sensitivity Analysis) with four levels of social distancing interventions that would affect the contact rate and therefore the background SARS-CoV-2 transmission, such that, in combination with the assumed level of pre-existing immunity, resulted in effective reproductive numbers (defined as the average number of secondary cases per infectious case in a population made up of both susceptible and non-susceptible hosts) of $R_{eff} = 1.1$ (observed in WA state in January 2021 [25]), 1.3, 1.5 and 2.4, respectively, at the beginning of our simulations (table 2, Fig. S2 and SM).”

Figure 2: It would be very helpful for the reader to be able to also see the predictions for other calibration metrics (deaths, healthcare burden), even though the campaign designs were optimised to reduce number of infections. This could be done with extra rows of panels, or as supplement to which can be referred in the figure caption.

We added an additional supplemental figure (Fig. S5) with the other metrics.

Discussion:

The values of R_t mentioned for moderate and high transmission don't match the range of values reported in the results section.

Thank you for catching the typo! Values of R_t have been modified.

Please discuss more about the potential impact of changes in risk behaviour in the population as vaccination is rolled out due to perceived declining risk. Sure, maintaining social distancing is paramount. Will people do it, though? Probably not as well as we would like, especially the “high transmission” groups. So, based on the findings, which of the examined strategies should be expected to be the most robust to lax social distancing? Please also discuss the impact of potentially lower willingness to take up vaccination among younger age groups, and in particular, the high-transmission age group.

These are very interesting points. Because we currently don't have any data supporting risk disinhibition (stratified by age, occupation, etc) at this point we didn't want to venture into the robustness of different strategies to risk disinhibition, but we have highlighted this as a consideration to be made by policy makers. We added a sentence in the discussion regarding this. It now reads:

“In practice, other factors need to be considered when allocating vaccine. These include differences in risk behavior, perceived risk and willingness to be vaccinated amongst different age groups, ethical and logistical considerations.”

Please highlight or discuss the consequence of using “deaths” as an optimisation metrics instead of “years of life lost” (which would favour preventing deaths in younger individuals).

We chose to minimize deaths as we believe that each human life should be treated equally, regardless of age. Throughout the world, this has been the prevailing prioritization. We acknowledge that there are other ethical considerations and that YYL could also be considered, as many other groups have done. We have added a sentence now in the discussion that reads as follows:

“Our results suggest that the optimal use of vaccine depends on the metric minimized. These metrics were chosen based on the prevailing scientific discussions surrounding COVID-19 vaccine prioritization [50, 51]. In particular, we chose to optimize cumulative number of deaths as opposed to years of life lost (YLL), and this resulted in attributing more weight to vaccinating the oldest age groups, which are at greatest risk of death. Other metrics, like YYL would give rise to different optimal strategies, that may favor younger age groups.”

Please highlight or discuss the relevance of long-term consequences of Covid-19, and how this may be derived from this work and how that might change optimal designs.

To the best of our knowledge, there is very little information regarding the long-term consequences of COVID-19, although research is undergoing and there is a big push to further advance our knowledge. Once more information is available (who is more affected, why, how long, etc) we will happily include this in our work, we have added a sentence in the discussion addressing this point. It reads as follows:

“Furthermore, new evidence is emerging about the long-term consequences of COVID-19 disease. This might be an important factor to incorporate in the metrics of disease burden in future work.”

Reviewer #2 (Remarks to the Author):

See attached Word file.

Reviewer #3 (Remarks to the Author):

This is an interesting manuscript dealing with the important issue of optimizing vaccine allocation.

I have some minor comments the authors might like to address:

p2: I am surprized you have not included your own paper from 2015 perhaps being the first to address one vs two vaccine responses.

Our 2015 paper is included, but in the discussion section. This is because in the intro we did not include any refs to modeling papers.

p4: I did not check the S.M: but I strongly encourage that you explicitly give the contact matrix between different age groups

Given that there are 16 age groups in the model and we have 4 scenarios that result in different contact matrices, we thought it would be difficult to explicitly give the contact matrices. Instead, we added a figure that shows the contact matrices used and how they are reduced for each scenario considered (Fig. S2) and referred to this figure, as well as the table where these matrices are defined, in the main text. We believe that this, in addition to the description given in SM will give the reader enough information to understand how these contact matrices were used.

p6: Why do you end at 50% receiving one dose? The question remains interesting even when having vaccines for e.g. 120%

We agree with the reviewer that having even more vaccine is a very interesting problem. In fact, we did the analysis for 100% of the population when considering a faster vaccination rate (300K /week , fig. S15). We concentrated at 50% because that reflected vaccinating steady at 150K/week, which at the time, given the available information regarding vaccine availability and government's capability to carry vaccination over the first 6 months, seemed to be the more realistic scenario.

p7: Do you really know it is optimal (mathematically)? Perhaps a few words about the algorithm in main text or SM.

No, we don't. This is stated in the SM: "We used a heuristic method because it is fast and using a method that returns a guarantee of optimality is very difficult for complex black-box objective functions. We took a very large sample of the entire feasible set to ensure a thorough search to increase the chance that our solution is nearly globally optimal. Furthermore, the sensitivity analysis shows that the optimal solutions we found are robust to changes in parameters and, even though we cannot guarantee they are globally optimal, our solutions either coincide with or outperform the pro-rata and high-risk strategies, so they are relatively optimal in that sense."

p7: How is parameter uncertainty expressed? Give specific reference to SM.

In order to make it more clear how parameter uncertainty was analyzed, we changed the following sentence in the main text:

"To assess parameter uncertainty, for each of the strategies compared and for each outcome, we ran the model with 1,000 different parameter sets representing the uncertainty surrounding those parameters we believe would be more likely to affect our results, and removed top and bottom 2.5% of the simulations to calculate uncertainty intervals (denoted below as 95% UI) reflecting the uncertainty in the outcomes arising from uncertainty in the parameter estimates (Uncertainty analysis, SM)."

p7, Results: I am surprised that the optimal strategy often vaccinates in all age groups. I would have expected filling one age group with at least one dose before going to another the next age group. Is the explanation to this that much of the contact mixing is on the diagonal? Please comment more in text. Also, why is the optimal fractions more or less equal between age groups when SDE is high? Comment.

The pro-rata strategy, in which all adult groups are vaccinated more or less in equal fractions, was chosen as a strategy that mimics an allocation in which all adults are eligible to be vaccinated and we assume that all age groups are equally likely to be vaccinated. We included this strategy because it is likely to be seen in some countries and might be considered "fair". In fact, in a similar vein, vaccine allocation across states in the US is done in a pro-rata fashion, where states are being allocated vaccine proportional to their overall population.

Because we included this strategy in our set of initial conditions for the optimization, it comes back as optimal under some scenarios. In particular, for high SDE, we expect the optimal allocation

strategy to allocate most of the vaccine in single dose campaigns and, under low viral transmission, the optimal allocation strategy is “balancing” doses between the groups that are responsible for most of the transmission, and those that account for most of the deaths.

We have added a sentence in the results to highlight what the pro-rata strategy represents that reads as follows:

“this strategy models an allocation in which all adults are eligible to be vaccinated and we assumed that all age groups are equally likely to be vaccinated”

Discussion: policies which recommend to vaccinate fractions in all age-groups are most likely hard to implement practically. Comment.

We agree with the reviewer. In reality, only fractions of each age group are likely to be vaccinated due to practical reasons and vaccine hesitancy. We believe that our results can provide guidance in the order in which one should vaccinate. In the discussion, we acknowledged the difficulty of implementing the optimal strategies, it reads:

“While some of the optimal allocation strategies may be difficult to implement, our results can be used to guide the development of mixed vaccination strategies, where some subpopulations receive one dose and others receive two doses, thereby achieving a balance between rapid coverage and full protection of those most at risk of severe disease and death.”

p17, mid-paragraph. There are of course also arguments for going for two vaccines. Even if these are more obvious I would state them, e.g.: vaccine guarantee, a second dose much later may not have same overall effect.

“We added the following sentence to the discussion:

However, strictly following the two-dose vaccination schedule might also be important, as it is uncertain the effectiveness of a second dose given beyond the 3 week- period tested in clinical trials, especially in the context of emerging variants [50].”

p18: a deterministic system defined by rates implicitly imply exponential durations in states. Don't change model, but acknowledge this.

We added this to our limitation section.

p18, regarding individual with anti-bodies. To me it is obvious that when having limited vaccine supply these individuals should not be vaccinated at all. Did you address this?

While we agree that when having limited supply of vaccine, previously infected individuals might not be the best candidates for vaccination, we did not address this in this paper. We consulted with experts in the implementation world (personal communication) and we were told that in general, we don't want to put barriers for people to get vaccinated (implementing a policy where only non-infected individuals would be more costly and lengthy, as people would have to be tested and get results before being eligible to be vaccinated), so that governments would tend not to request additional tests prior to vaccination. This has been largely validated by the current vaccination programmes throughout the world. To the best of our knowledge, governments are not requiring anything related to prior exposure to be eligible to get a vaccine. Moreover, recent work demonstrates a robust boosting of antibody levels following a single shot in persons with previous infection and boosting with vaccines might provide additional benefits against different variants. In our model, we allowed for people who have had an asymptomatic infection to be vaccinated but that has no further effect, and this is mentioned in the limitation section in the discussion.

p19, line f4: peak hospitalizations are overestimated and happen too early.

We modified this sentence in the text to add the pace of hospitalizations and now reads
“it is possible that our peak hospitalizations are overestimated and are occurring at a faster pace than in reality.”

p19, middle. I don't at all understand comment that you only consider a fixed prioritization scheme. I thought the whole idea was that you compared different orders?

We did not compare different orders, but different age groups to be vaccinated. Once a particular allocation strategy is chosen, we follow the same vaccination implementation, where we always start with the older age group for that particular allocation, and vaccinate in decreasing order. So for example, if a particular strategy calls to vaccinate adults 50-65 and those aged 20-50 we will always start by vaccinating those 50-65 and then those 20-50.

We have added now a sentence in the results section that explains this and refers to the SM, where we have given a detailed explanation. The sentence reads as follows:

“In order to perform the optimization, we implemented the vaccination campaigns in an identical way for all the allocation strategies considered: the older age group in a particular strategy is vaccinated first and we moved sequentially in decreasing order across the vaccine groups (full details SM).”

Review of NCOMMS-21-08478

Optimizing vaccine allocation for COVID-19 vaccines: potential role of single dose vaccine.

Comments:

This paper tackles a very important and relevant problem playing out literally today in terms of what is the role for a single dose vaccine regimen versus a two-dose regimen and can simulation-based modeling reveal optimal allocation strategies that can help guide actual practice on the ground. The research team conducted a very comprehensive set of analyses taking into account many potential factors using a deterministic modeling framework. They then conducted an extensive sensitivity analysis whereby certain assumptions were varied and the results compared with respect to five important metrics of disease burden. Models are constructed using Washington state as a framework for inferences. Important conclusions from the simulations are reached and an extensive discussion focusing very much on study limitations is given (which is to be very much lauded). In fact lots of room is dedicated to fleshing out these limitations and how things could be improved.

The modeling descriptions in the appendix look to be complete and the simulations are summarized nicely using both graphical and numerical approaches. Overall, this is a very nice

contribution to the research literature. I do have a few specific comments which would improve the paper further:

Since there are multiple variants circulating in the US now, it seems to me that rather than varying a single R_{eff} as a tuning parameter, that a mixture of R_{eff} s should be used each reflecting a different variant.

Other public health measures are still be encouraged and are being adapted to varying degrees by state or local areas. Building this kind of adoption into the modeling framework can be useful since the vaccine program efficacy will also interact with these other public health measures like masking and social distancing.

We thank the reviewer for this comment, we realize that our description of how R_{eff} was computed was not clear. In our model, R_{eff} is a combination of a baseline R_0 , the level of pre-existing immunity in the population, and the social distancing interventions being adopted. We have changed the wording in the results section to make this clear, it reads now as follows:

“We considered a baseline $R_0 = 3$ (alternative scenario $R_0 = 4$, Sensitivity Analysis) with four levels of social distancing interventions that would affect the background SARS-CoV-2 transmission, such that, in combination with the assumed level of pre-existing immunity, resulted in effective reproductive numbers (defined as the average number of secondary cases per infectious case in a population made up of both susceptible and non-susceptible hosts) of $R_{eff} = 1.1$ (observed in WA state in January 2021 [25]), 1.3, 1.5 and 2.4, respectively, at the beginning of our simulations (table 2, Fig. S2 and SM).”

In addition, we have added a subsection in the Sensitivity analysis section where we repeated the results assuming increased baseline transmission, with $R_0 = 4$.

Multiple vaccines are being used at the same time – some with differing efficacies at preventing infection. This should be embedded into the modeling structure.

Even though the focus is in the Washington state area, the patchwork process by which the US has rolled out programs could also be simulated. This would mean allowing for more local structure in the modeling.

We have added a section in the Sensitivity analysis where we repeated the work assuming a much lower vaccine efficacy. However, this model doesn't have the ability of representing different vaccine efficacies and adding this would result in a completely different and far more complicated optimization problem (that we will tackle in future work). We have noted this as a limitation of our work.

Everything is done using deterministic models. Some small discussion around what might be expected if stochastic variations of the models were used instead.

We have discussed and expanded the used of deterministic models in our limitations paragraph and currently reads as follows:

“Further, deterministic models implicitly result in exponential durations for each disease state, and they can overestimate infection dynamics. While comparisons with and without vaccination strategies would not be affected by this, it is possible that our peak hospitalizations are overestimated and are occurring at a faster pace than in reality.”

REVIEWERS' COMMENTS

Reviewer #1 (Remarks to the Author):

The authors have adequately addressed the points I raised; I have no further feedback to give.

Reviewer #2 (Remarks to the Author):

I feel this revision has been done nicely and all questions brought up by the reviewers adequately addressed.